# AgentRefine: Enhancing Agent Generalization through Refinement Tuning

**Dayuan Fu**[1]*, **Keqing He**[2]*, **Yejie Wang**[1]*, **Wentao Hong**[1], **Zhuoma Gongque**[1], **Weihao Zeng**[1],
**Wei Wang**[2], **Jingang Wang**[2], **Xunliang Cai**[2], **Weiran Xu**[1] †
[1]Beijing University of Posts and Telecommunications, Beijing, China
[2]Meituan, Beijing, China

## Abstract

Large Language Model (LLM) based agents have proved their ability to perform complex tasks like humans. However, there is still a large gap between open-sourced LLMs and commercial models like the GPT series. In this paper, we focus on improving the agent generalization capabilities of LLMs via instruction tuning. We first observe that the existing agent training corpus exhibits satisfactory results on held-in evaluation sets but fails to generalize to held-out sets. These agent-tuning works face severe formatting errors and are frequently stuck in the same mistake for a long while. We analyze that the poor generalization ability comes from overfitting to several manual agent environments and a lack of adaptation to new situations. They struggle with the wrong action steps and can not learn from the experience but just memorize existing observation-action relations. Inspired by the insight, we propose a novel AgentRefine framework for agent-tuning. The core idea is to enable the model to learn to correct its mistakes via observation in the trajectory. Specifically, we propose an agent synthesis framework to encompass a diverse array of environments and tasks and prompt a strong LLM to refine its error action according to the environment feedback. AgentRefine significantly outperforms state-of-the-art agent-tuning work in terms of generalization ability on diverse agent tasks. It also has better robustness facing perturbation and can generate diversified thought in inference. Our findings establish the correlation between agent generalization and self-refinement and provide a new paradigm for future research.

## 1 Introduction

Language agents (Mialon et al., 2023; Sumers et al., 2023), which harness the powerful capabilities of large language models (LLMs) to perceive environments, make decisions, and take actions, have emerged as an effective solution to complex real-world problems. Plenty of agent projects such as AutoGPT (Sig), GPT-Engineer (gpt), and BabyAGI (yoh) have employed LLMs as the core controllers, showing potential for practical applications. Both prompt engineering (Yao et al., 2022; Fu et al., 2024; Zhao et al., 2024) and framework practice (Yao et al., 2024; Shinn et al., 2024) have been proposed to enhance the agent capability of top-tier commercial LLMs like GPT-4. Recently, open-sourced LLMs (Dubey et al., 2024; Jiang et al., 2023) are emerging as effective alternatives to GPT models and show promising results.

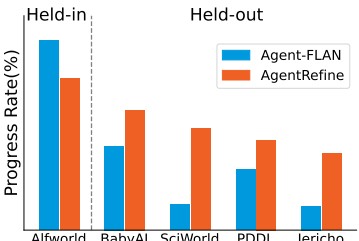

Figure 1: Overall progress score among 5 tasks. Agent-FLAN has been trained on Held-in task.

Many efforts have been made to enhance the agent capability of open-sourced LLMs via finetuning. Deng et al. (2024); Qin et al. (2023) carefully define single task schema and collect agent data for

---

*Equal contribution. Emails: `fdy@bupt.edu.cn`, Code: https://github.com/Fu-Dayuan/AgentRefine
†Corresponding authors.

specific vertical fields. Further, Zeng et al. (2023); Chen et al. (2024); Hu et al. (2024) extend to diverse agent tasks and cover high-quality Chain-of-Thought (CoT) rationale (Yao et al., 2022) to enhance the agent performance on unseen tasks. Although these works achieve admirable performance on held-in agent tasks where the collected training data share the same environment, their generalizability to more held-out sets is poor (shown in Figure 1). To solve the generalization issue of agent-tuning, (Zeng et al., 2023; Chen et al., 2024) mix general alignment data, ShareGPT (Chiang et al., 2023) with their agent data. They conclude that the general capabilities of LLMs are necessary for the generalization of agent tasks and training solely on agent data always leads to a decline in held-out agent performance.

In this work, we revisit the hypothesis that training solely on agent data can't generalize to new environments and delve into the reasons behind agent capability generalization. We first investigate the errors of the existing agent-tuning work in the new agent environments and most of them are formatting errors, illogical reasoning, and duplicated generation. While the integration of general data ratios can partially mitigate these errors, we find current agent models struggle with the same mistake and repeat erroneous actions, even when the environment provides explicit negative feedback. Inspired by (Shinn et al., 2024; Madaan et al., 2024), we connect the generalization of agent capability with self-refinement (Madaan et al., 2024) according to the feedback signals from the agent environment. We argue a good agent should recognize its mistakes and refine the previous actions by interacting with the environment. The self-refinement ability enables the agent to learn from its mistakes, avoiding getting trapped in a specific predicament, and allows it to discover the correct sequence of actions through reasonable exploration.

Expanding on the aforementioned insight, our objective is to develop generalized agent-tuning data and establish the correlation between agent generalization and self-refinement. To this end, we first propose an agent synthesis framework to encompass a diverse array of environments and tasks drawing upon extensive human persona data (Chan et al., 2024) that reflects various professional roles and personal interests. The diversity of agent environments prevents the model from overfitting to a single scenario. Then for each generated agent environment and corresponding task, we ask a strong LLM to simulate a multi-turn interaction. After generating each turn, we use a verifier to detect whether it contains format or logical errors. We keep the error turn and prompt LLM to refine its action according to the observation. The final agent data will undergo self-refinement processes and ultimately lead to a correct result. We find that agent-tuning on the self-refinement data, which we call `Refinement Tuning`, enhances the agent to explore more viable actions while meeting bad situations, thereby resulting in better generalization to new agent environments.

In this paper, we present AgentRefine, which investigates the self-refinement in agent-tuning to enhance agent generalization. We perform refinement tuning using our synthesis data on the LLaMA3 (Dubey et al., 2024) and Mistral-v0.3 (Jiang et al., 2023). Our experiments in terms of five agent evaluation tasks demonstrate that AgentRefine significantly outperforms state-of-the-art agent-tuning work. The key findings are summarized as follows:

- While existing agent-tuning work improve held-in agent performance, they hardly generalize the ability to new agent tasks. In contrast, our AgentRefine does not depend on memorizing training trajectories but learns to self-refine its mistakes and explore more actions and reasonable paths.

- Our experiments demonstrate that agent-tuning on normal trajectories performs poorly to the small perturbation of agent environments, like the action description. Refinement tuning exhibits greater robustness to environmental changes.

- Further analysis indicates the diversity of agent environments and thoughts contributes to refinement tuning.

## 2 RETHINK THE GENERALIZATION OF AGENT-TUNING

*Current agent-tuning works lack generalization to new agent tasks.* Figure 1 compares the performance between held-in and held-out agent tasks, where Agent-FLAN utilizes the Alfworld environment to gather training data and subsequently makes direct predictions for the held-out tasks. We observe a clear performance drop between the two settings.

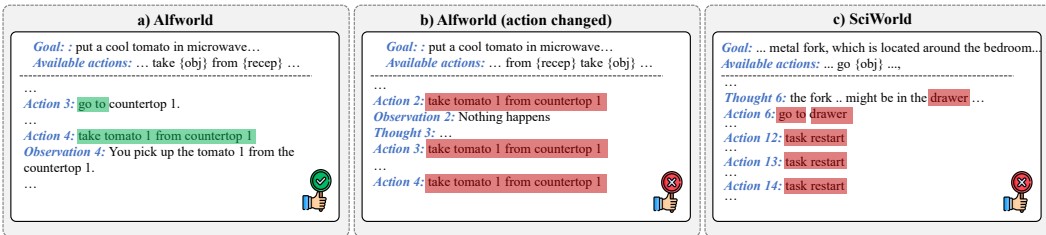

Figure 2: Example of parameter memorization in Agent-FLAN.

*Memorizing true trajectories leads to overfitting.* To further figure out the reason behind the poor generalization, we employ a study on the robustness of Agent-FLAN. Figure 2 displays the different output results in three evaluation settings where (a) denotes the original output in the held-in Alfworld task, (b) represents the modified Alfworld task with only reordering the action description, and (c) means the held-out SciWorld task. Agent-FLAN fits well into the held-in agent environment but fails to recognize subtle perturbations or handle new tasks (§4.3). Moreover, we analyze the bad cases of existing agent-tuning work in the held-out tasks and observe that once the model outputs an

error action, the entire process will be stuck in the same error mode for a while, regardless of the observation (§7). These experimental results indicate that traditional approaches merely memorize the correct trajectory information, fundamentally leading to a lack of generalization capability.

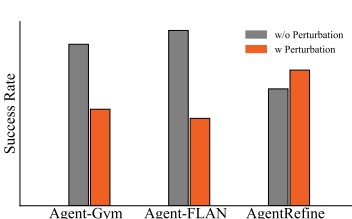

Figure 3: The success rate variation via perturbation

*Not memorize but self-refine.* Inspired by recent work (Shinn et al., 2024; Madaan et al., 2024), we connect the generalization of agent capability with self-refinement based on environment feedback. We hypothesize that self-refinement ability enables the agent to learn from its mistakes and discover the correct sequence of actions through reasonable exploration (§4.2).

## 3 METHODOLOGY

### 3.1 DATA CONSTRUCTION

Inspired by the Tabletop Role-playing game (TRPG), AgentRefine data's construction process can be divided into three parts: script generation, trajectory generation, and verification, as shown in Figure 4. The script generation requires the LLM to generate a script with the environment, tasks, and available actions based on the persona. In the trajectory generation phase, the LLM is required to simultaneously play the roles of both Dungeon Master (DM) and player to generate multi-turn agent data **containing errors and refine steps** based on the script. The verification will verify the script and trajectory, giving LLM the mistake it has made within a given persona and the LLM will regenerate the script/trajectory based on the verifier's response.

**Script Generation** We first sample a persona $p_i$ from diverse personas (Chan et al., 2024), and prompt the LLM to generate a script with the environment, tasks, and available actions based on $p_i$. The environment will include locations, items, and player information that may appear in the interaction. To assist the LLM in understanding the environment, we prompt the LLM to display the hierarchical relationships between locations/items in JSON format. We also require the LLM to generate some interfering locations/items, to ensure that some erroneous steps are likely to occur during trajectory generation. After generating the environment, the LLM will generate a clear and specific task. Finally, the LLM will generate a series of available actions. For each action, we require the LLM to generate an action name, validation code (a regular expression), and valid parameters. The structure of the script can be seen in Appendix L.

**Trajectory Generation** Given a script, the LLM can simulate multi-turn interactions between the DM and the player within one call. Specifically, the DM's turn is divided into three stages: thinking,

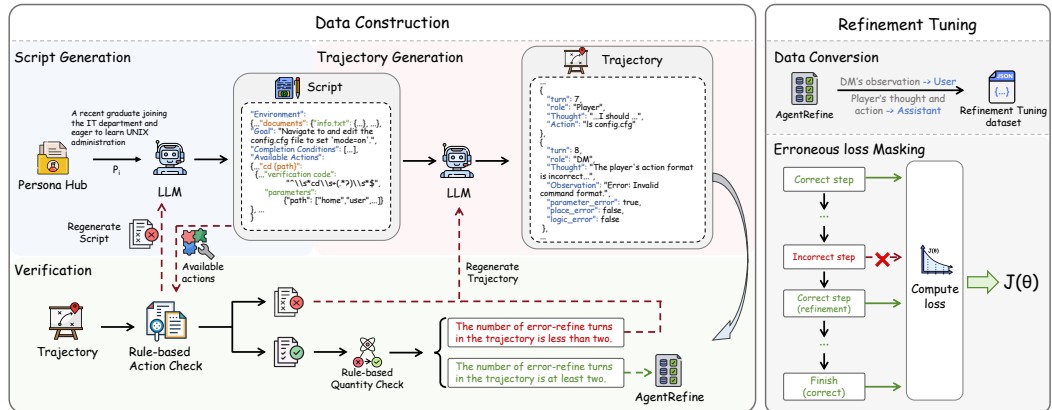

Figure 4: The pipeline of AgentRefine data generation and refinement tuning.

observing, and evaluating. In the thinking stage, we require the LLM to evaluate the player's state and known information so far and analyze the observations the player can obtain based on the last action. The observing stage will provide the observations the player can obtain, while in the evaluating stage, the DM will assess whether the player's last action contains parameter errors, logical errors, and location errors (act in the wrong place). The player's turn is similar to ReAct, requiring the LLM to analyze the current state through thought and then propose an action. The structure of the trajectory can be found in Appendix M.

**Verification** The verifier will check both the script and the trajectory. In script part, to ensure the validity of the action names, we apply the validation code on the action names and only save the script if all actions pass the validation [1]. In the trajectory part, if the generated trajectory has: (1) JSON format error at a certain turn $t$, (2) The task is not completed in the final turn $t - 1$ (3) In the player's $t$ turn its action can not match any validation code with corresponding parameters and the DM does not provide a parameters error in turn $t + 1$, we will save all previous turns up to $t - 1$ and prompt the LLM to continue generating. If the DM evaluates that the task is completed but the number of error-refine turns in the trajectory is less than two, we will provide all turns to the LLM and require it to regenerate the trajectory from the beginning. Detailed verification steps can be seen in Appendix O.

## 3.2 GENERATION SETUP

We use gpt-4o-2024-05-13 to generate the script and trajectory. We will save all trajectories that can pass verification in 4 LLM calls (including script generation and trajectory generation). We primarily adopt the 1-shot trajectory example approach in trajectory generation and the 3-shot script examples in script generation to help LLM follow the format and give a diversified result. In Appendix 5, we use deepseek-v2.5 (Liu et al., 2024) as the open-source LLM to generate the script and trajectory.

## 3.3 REFINEMENT TUNING

After generating the complete trajectory, we convert the trajectory into a Refinement Tuning dataset $D_{RT}$, specifically, the user turn is the DM's observation, while the assistant turn is the Player's thought and action, in ReAct (Yao et al., 2022) format. To prevent interference from error turns generated by the LLM, we changed the loss function $J(\theta)$, as shown in Equation 1 where $N_x$ is the total turn number of a given data $x$, $T_j, A_j, O_j$ is the thought, action, and observation in turn $j$. If $A_j$ is correct $\mathbf{1}(A_j) = 1$ else $\mathbf{1}(A_j) = 0$.

$$J(\theta) = \mathbb{E}_{x \sim D_{RT}} \left( \sum_{i=1}^{N_x} \log \left( \pi_\theta \left( T_i, A_i | I, \{T_j, A_j, O_j\}_{j=0,\dots,i-1} \right) \mathbf{1}(A_j) \right) \right) \tag{1}$$

---

[1]Due to the near-infinite parameter space of actions in virtual environments such as code editing, answering, and searching, these actions will not be verified in both script generation and trajectory generation

## 4 EXPERIMENTS

### 4.1 EXPERIMENT SETUP

**Training** We use the LLaMA3-base series models (Dubey et al., 2024) for most of our experiments. For mistral (Jiang et al., 2023), we use mistral-v0.3. We applied the original llama3 (or mistral)'s multi-turn chat template. We use LLaMA-Factory (Zheng et al., 2024) to train our models. The training hyperparameter details can be seen in Appendix D.

**Tasks** We select 5 tasks: SciWorld (Wang et al., 2022), Alfworld (Shridhar et al., 2020), BabyAI (Chevalier-Boisvert et al., 2018), PDDL (Vallati et al., 2015), and Jericho (Hausknecht et al., 2020), all of them are testing models' decision-making ability. We use the AgentBoard (Ma et al., 2024) framework for experiments, this framework can determine whether the agent has completed all tasks (success rate) and whether the agent has reached key nodes (progress rate). The Held-in task refers to Alfworld, while the Held-out tasks are the results obtained by the weighted average of other tasks based on AgentBoard (Ma et al., 2024) We change AgentBoard's prompts from Act-only to ReAct and the historical thought, action, and observation will be transformed into the chat format instead of plaintext. We adjusted the example prompts on Llama-3-8B-Instruct and never changed them during this work. (except §4.3). The max turn is 30 for all these tasks in inference. To further prove AgentRefine's generalization, followed by the choice in ReAct (Yao et al., 2022), We choose a reasoning task HotpotQA (Yang et al., 2018) in the ablation experiment. We use Wikipedia search in LATS (Zhou et al., 2023) as the environment, randomly sample 300 questions from HotpotQA, and test the exact match (EM) and F1 score of those methods. The max turn is 8 for HotpotQA task in inference. It should be emphasized that we will only use environment feedback in the inference and we will not use GPT4's judgement as the feedback.

**Baseline** For the close-source model, we choose GPT-4o (gpt-4o-2024-05-13) and GPT4o-mini (gpt-4o-mini-2024-07-18). For the open source model, we choose Meta-Llama-3-8B-Instruct, Meta-Llama-3-70B-Instruct, and Mistral-7B-Instruct-v0.3. For fine-tuned mode, we choose Agent-FLAN (Chen et al., 2024), AgentGym (Xi et al., 2024), and AgentGen (Hu et al., 2024) as the baseline. They are all trying to solve the agent generalization problem. Agent-FLAN is an improvement of AgentTunning (Zeng et al., 2023), focusing on training "thought" in ReAct. AgentGym uses lots of environments to ensure generalization and AgentGen uses LIMA (Zhou et al., 2024) to synthesize diversified agent-tuning data. Agent-FLAN includes Alfworld in its training set. AgentGym includes Alfworld, BabyAI, and SciWorld in its training set. These datasets will be seen as Held-in test tasks for the corresponding method. Since Agent-FLAN and AgentGym's original model is LLaMA2-Chat, for a fair comparison, we reproduce them under LLaMA3 and Mistral. Since AgentGym has not open sourced, we only report the result in (Hu et al., 2024)

### 4.2 MAIN RESULTS

Table 1 shows the performance comparison of AgentRefine and other methods across different families and sizes. It is important to emphasize that some methods sample training data in the same environment as the task; in such cases, we consider this task for these methods to be held-in. We identify the held-in metrics for each method with an underscore. It can be observed that compared to other agent works, our method shows significant advantages in held-out tasks. For example, it leads Agent-FLAN by 13.3% in Sciworld Success Rate. Notably, in some tasks, AgentRefine can even match the performance of the GPT-4o series. This demonstrates the strong generalization capability of AgentRefine. We also observe that AgentRefine can not outperform held-in training methods. However, in § 4.3, we will demonstrate that these held-in methods simply memorize the mapping between observation and action, and a very small perturbation can render these methods ineffective. Furthermore, we also notice that LLaMA-3-8B-Instruct exhibits very strong performance in many tasks. We attribute this to its extensive use of Alignment data and additional RL training. In subsequent experiments, we also mix alignment data and AgentRefine and achieve further gains.

**Effect of Refinement Tuning** To further investigate the effectiveness of Refinement Tuning, we mask the loss of refinement trajectory tokens. Table 2 shows that after masking the refinement, the model's performance over 5 tasks drops dramatically. For instance, there is approximately 43% performance drop in Sciworld which, to some extent, reflects the necessity of Refinement Tuning for Agent tasks. we also re-generated a training set without error and refinement trajectories, which

| Method | Alfworld | | BabyAI | | SciWorld | | PDDL | | Jericho | |
|---|---|---|---|---|---|---|---|---|---|---|
| | Success | Progress | Success | Progress | Success | Progress | Success | Progress | Success | Progress |
| *GPT Series* | | | | | | | | | | |
| GPT-4o | 66.4 | 79.9 | 48.2 | 64.1 | 40 | 76.9 | 61.7 | 69.8 | 10.0 | 34.0 |
| GPT-4o-mini | 37.3 | 65.0 | 36.6 | 51.9 | 23.3 | 49.8 | 25.0 | 49.1 | 10.0 | 28.5 |
| *LLaMA-3-8B Series* | | | | | | | | | | |
| LLaMA-3-8B-Instruct | 22.4 | 46.1 | 45.5 | 56.5 | 7.8 | 41.1 | 10.0 | 38.4 | 0.0 | 24.3 |
| AgentGen | 29.1 | 47.6 | 20.5 | 35.0 | - | - | 11.7 | 23.0 | - | - |
| AgentGym | 61.9 | 76.9 | 47.3 | 61.4 | 18.9 | 47.5 | 1.7 | 16.6 | 0.0 | 12.9 |
| Agent-FLAN | 67.2 | 79.7 | 25.0 | 35.3 | 1.1 | 10.9 | 8.3 | 25.5 | 0.0 | 10.1 |
| AgentRefine | 44.8 | 63.8 | 37.5 | 50.4 | 14.4 | 42.6 | 16.6 | 37.8 | 10.0 | 32.3 |
| *Mistral Series* | | | | | | | | | | |
| Mistral-7B-Instruct-v0.3 | 12.4 | 35.9 | 36.6 | 45.8 | 6.7 | 24.7 | 13.3 | 27.8 | 0.0 | 17.3 |
| AgentGym | 76.9 | 86.7 | 40.2 | 56.3 | 15.6 | 48.3 | 1.7 | 7.3 | 0.0 | 13.0 |
| Agent-FLAN | 77.6 | 87.6 | 15.2 | 21.0 | 0 | 6.7 | 0 | 3.2 | 0.0 | 0.7 |
| AgentRefine | 51.4 | 68.8 | 25.9 | 42.4 | 4.4 | 22.4 | 11.7 | 32.8 | 5.0 | 28.8 |
| *LLaMA-3-70B Series* | | | | | | | | | | |
| LLaMA-3-70B-Instruct | 67.2 | 75.2 | 48.2 | 61.8 | 42.2 | 75.4 | 55.0 | 79.8 | 25.0 | 46.4 |
| Agent-FLAN | 80.5 | 86.8 | 32.1 | 41.2 | 5.5 | 16.4 | 25.0 | 53.7 | 0.0 | 13.6 |
| AgentRefine | 67.2 | 72.1 | 44.6 | 59.7 | 17.7 | 46.4 | 38.3 | 58.6 | 15.0 | 37.2 |

Table 1: Main Results. The underlined text indicates that the training data is sampled in the same environment as the task and is considered as held-in evaluation. We use the original result in Agent-Gen and reproduce AgentGym and Agent-FLAN's results.

| Method | Alfworld | | BabyAI | | SciWorld | | PDDL | | Jericho | |
|---|---|---|---|---|---|---|---|---|---|---|
| | Success | Progress | Success | Progress | Success | Progress | Success | Progress | Success | Progress |
| AgentRefine | 48.5 | 61.5 | 37.1 | 51.7 | 7.7 | 33.1 | 21.7 | 37.4 | 5.0 | 26.2 |
| - w/o refinement loss | 40.3 | 58.8 | 34.8 | 45.6 | 4.4 | 22.7 | 20.0 | 37.4 | 0.0 | 16.1 |
| - w/o refinement data | 49.3 | 65.2 | 30.4 | 43.1 | 5.5 | 21.3 | 11.7 | 32.5 | 0.0 | 13.8 |
| - w erroneous loss | 29.9 | 43.9 | 23.2 | 31.6 | 3.3 | 19.0 | 8.3 | 28.3 | 5.0 | 18.4 |

Table 2: Ablation study of Refinement Tuning. This experiment is in the data size of 8000.

completely eliminates the impact of Refinement Tuning. From Table 2, we can observe that the model trained on data without refinement trajectories experiences a similar magnitude of performance drop across all tasks.

In our proposed Refinement Tuning, we mask the loss of erroneous turn tokens to prevent the model from learning incorrect thought processes. To verify whether this process is necessary, we train a model learning all assistant turn tokens on the same data. Table 2 shows that the model learned erroneous tokens results in very adverse consequences, with nearly a 75% drop in Sciworld. This conclusion is contrary to (Ye et al., 2024). In fact, we find that the model's performance on these tasks can continue to drop to a low level with the continued learning of data with erroneous trajectories. We believe that at least for agent Refinement Tuning, eliminating the loss of erroneous turns is crucial. Otherwise, models will learn incorrect reasoning processes, leading to poor performance on held-out tasks.

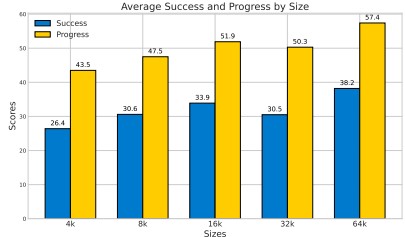

Figure 5: The model's performance as the AgentRefine train data scales up.

**Scaling AgentRefine** We experiment and analyze the relationship between the data size of the AgentRefine training set and model performance, with the results shown in Figure 5. From the results, we can observe that the model demonstrates significant gains in performance as the data size increases from 4k to 64k, which illustrates the effectiveness of the AgentRefine data.

## 4.3 ROBUSTNESS ANALYSIS

Previous work has extensively trained on held-in tasks but shows poor performance on held-out tasks. One possible reason is that models simply memorize the key-value pairs between observation

| Model | Alfworld | | Perturbation 1 | | Perturbation 2 | | Perturbation 3 | | Perturbation 4 | | Perturbation 5 | | Average | | Std | |
|---|---|---|---|---|---|---|---|---|---|---|---|---|---|---|---|---|
| | Success | Progress | Success | Progress | Success | Progress | Success | Progress | Success | Progress | Success | Progress | Success | Progress | Success | Progress |
| LLaMA3-8B-Instruct | 22.4 | 46.1 | 23.1 | 45.6 | 24.6 | 45.0 | 17.9 | 45.1 | 17.9 | 45.1 | 22.4 | 46.1 | 21.4 | 45.5 | 2.68 | 0.47 |
| AgentGym | 61.9 | 76.9 | 29.1 | 59.2 | 49.2 | 65.3 | 32.8 | 53.9 | 38.8 | 48.2 | 5.9 | 28.7 | 36.3 | 55.4 | 19.97 | 16.66 |
| Agent-FLAN | 67.2 | 79.7 | 21.6 | 58.8 | 51.4 | 71.3 | 27.6 | 53.5 | 52.2 | 67.9 | 1.5 | 19.7 | 36.9 | 58.5 | 21.98 | 22.53 |
| AgentRefine | 44.8 | 63.8 | 50.0 | 66.5 | 51.5 | 66.7 | 54.5 | 70.0 | 45.5 | 60.6 | 44.8 | 63.8 | 48.5 | 65.2 | 3.73 | 3.56 |

Table 3: Performance for different models across various perturbations.

and actions from training data, rather than learning to infer correct actions based on the task and observation. To test the hypothesis above, we conduct data perturbation experiments on a held-in task. Specifically, we select the Alfworld, which belongs to the held-in category for both AgentGym and Agent-FLAN. We perturb the candidate actions in Alfworld ensuring that the perturbed ones consist of different tokens (or token order) but express the same semantic information. The detail perturbation rules are shown in Appendix K.

Table 3 shows the experimental results. It can be observed that simple data perturbation leads to a significant performance drop on the original held-in task. For example, under the average score, AgentGym's Success Rate drops by 25.6%, while Agent-FLAN experiences an even more severe performance decline of 30.4%. Their standard deviation is close to 20%. In comparison, Our AgentRefine has a 3.7% increase in the average and low standard deviation, 3.73%, indicating that it learns decision-making capabilities rather than just simple memorization.

## 4.4 DIVERSITY ANALYSIS

**Thought Diversity** Figure 6 illustrates the distribution of chain-of-thought diversity across three agent datasets. We extracted the *thought* content from all ReAct rounds and vectorized them. We randomly sampled 8100 data from all *thoughts* and visualized them via dimensionality reduction using t-SNE (Van der Maaten & Hinton, 2008). Compared to Agent-FLAN and AgentGym, the data of AgentRefine are more widely distributed and numerous in Figure 6, indicating a higher diversity of *thoughts* in AgentRefine. This suggests that the AgentRefine data can better teach the model to think diversely, achieving a broader exploration space.

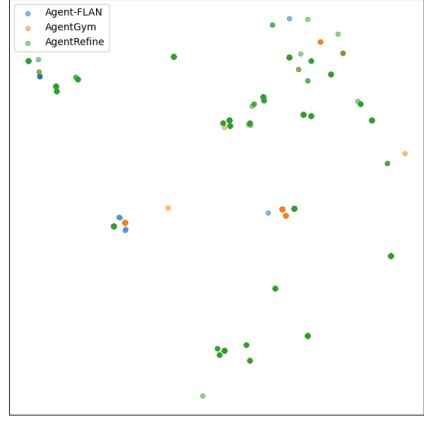

Figure 6: The t-SNE figure among Agent-FLAN, AgentGym, and AgentRefine's Thought.

**Environment Diversity** Figure 7 shows the similarity relationship between the AgentRefine environment and the test datasets. We randomly selected the instructions from 100 data (50 from AgentRefine and 10 from each test set) and removed the one-shot examples from the test sets. As shown in Figure 3, the similarity between the AgentRefine environment and the test environments is less than 0.5 (bottom left and top right sections), indicating a certain degree of difference between our environment and the test environments.

**Best-of-N** Table 4 presents the performance of the three agents on Best-of-N (BoN). We set the decoding temperature to 1, executed each target task ten times, and took the highest score as the progress rate. If there was at least one successful result among the ten executions, the success rate would be 1; otherwise, it would be 0. The results in Table 4 show that the BoN performance using any training data is always better than greedy, with the improvement of AgentRefine being particularly notable, averaging

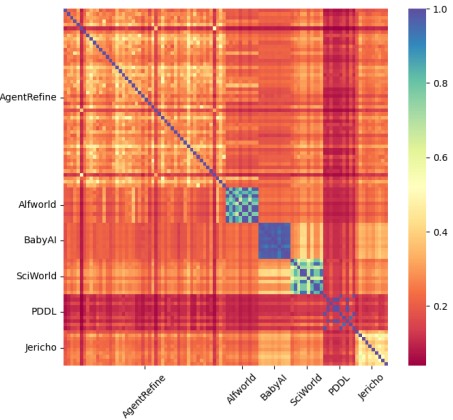

Figure 7: The similarity heatmap between different environments in 6 sources.

| Model | Alfworld | | BabyAI | | SciWorld | | PDDL | | Jericho | |
|---|---|---|---|---|---|---|---|---|---|---|
| | Success | Progress | Success | Progress | Success | Progress | Success | Progress | Success | Progress |
| AgentGym-greedy | 61.9 | 76.9 | 47.3 | 61.4 | 18.9 | 47.5 | 1.7 | 16.6 | 0.0 | 12.9 |
| AgentGym-BoN | 99.3 | 99.3 | 73.2 | 87.2 | 58.9 | 85.6 | 16.6 | 42.1 | 5.0 | 22.2 |
| Δ | 37.4 | 22.4 | 25.9 | 25.8 | 40.0 | 38.1 | 14.9 | 25.5 | 5.0 | 9.3 |
| Agent-FLAN-greedy | 67.2 | 79.7 | 25.0 | 35.3 | 1.1 | 10.9 | 8.3 | 25.5 | 0.0 | 10.1 |
| Agent-FLAN-BoN | 85.5 | 98.1 | 43.8 | 56.7 | 10.0 | 33.5 | 11.7 | 39.8 | 5.0 | 22.2 |
| Δ | 28.3 | 18.4 | 18.8 | 21.4 | 8.9 | 22.6 | 3.4 | 14.3 | 5.0 | 12.1 |
| AgentRefine-greedy | 44.8 | 63.8 | 37.5 | 50.4 | 14.4 | 42.6 | 16.6 | 37.8 | 10.0 | 32.3 |
| AgentRefine-BoN | 93.3 | 96.6 | 67.0 | 81.5 | 40.0 | 71.0 | 30.0 | 57.3 | 25 | 52.5 |
| Δ | 48.5 | 32.8 | 29.5 | 31.1 | 25.6 | 28.4 | 13.4 | 19.5 | 15.0 | 20.2 |

Table 4: Best-of-N results among three methods.

over 25%. The marked improvement of AgentRefine compared to the other two datasets is likely due to its higher diversity and quality of chain-of-thought. It also demonstrates that existing agent-tuning models have great potential. To gradually improve the model's performance, this result suggests that we should construct better reinforcement learning agent data towards generalization in future work.

## 5 SYNTHESIS FROM OPEN SOURCE MODEL

In the main experiment, we use GPT-4o to synthesize the AgentRefine data. In this chapter, we attempt to replace it with open-source models to complete the data synthesis process. Table 5 shows our results under 4000 training data. It can be observed that, compared to Agent-FLAN, which used GPT-4 for data synthesis, the AgentRefine data synthesized with the open-source model DeepSeek-v2.5 exhibits significant advantages on the held-out tasks. For example, it leads Agent-FLAN by 11.6% in the BabyAI Success Rate metric, further proving the advantages of AgentRefine. Additionally, we observe a noticeable gap between the data synthesized with DeepSeek and the data synthesized with GPT-4o. This indicates that using more capable models for data synthesis does indeed yield higher-quality training data and results in greater performance gains.

| Model | Alfworld | | BabyAI | | SciWorld | | PDDL | | Jericho | |
|---|---|---|---|---|---|---|---|---|---|---|
| | Success | Progress | Success | Progress | Success | Progress | Success | Progress | Success | Progress |
| Agent-FLAN | 67.2 | 79.7 | 25.0 | 35.3 | 1.1 | 10.9 | 8.3 | 25.5 | 0.0 | 10.1 |
| AgentRefine-DeepSeek | 32.0 | 44.2 | 36.6 | 48.1 | 2.2 | 21.6 | 16.6 | 36.7 | 5.0 | 29.0 |
| AgentRefine-GPT-4o | 36.6 | 55.9 | 33.9 | 44.1 | 11.1 | 31.4 | 18.3 | 37.9 | 10.0 | 28.8 |

Table 5: Performance on Different Synthesis Models, we synthesize 4000 data via deepseek-v2.5. The underlined text indicates that the training data is sampled in the same environment as the task and is considered as held-in evaluation

## 6 GENERLIZATION IN REASONING TASK

Figure 8 presents the results on the reasoning task, HotpotQA (Yang et al., 2018). The result shows that AgentRefine outperforms other methods on HotpotQA's EM and F1 metrics. It proves that AgentRefine's generalization still works on reasoning problems.

| Method | EM | F1 |
|---|---|---|
| LLaMA-3-8B-Instruct | 29.3 | 36.6 |
| AgentGym | 28.0 | 37.4 |
| Agent-FLAN | 24.6 | 32.4 |
| AgentRefine | 37.0 | 44.6 |

Figure 8: Model Performance on reasoning task, Hotpot QA.

## 7 CASE STUDY

Figure 9 presents examples of Agent-FLAN and AgentRefine in Jericho and Sciworld. The cases show that Refinement Tuning can enhance the diversity and quality of the model's thinking, which helps improve the model's exploration breadth and efficiency and avoid always getting stuck in loops in a new environment.

In Jericho, Agent-FLAN mistakenly believes it is not in the cell and attempts to *go to cell*. After failing, it chooses to *check valid actions*. Although *check valid actions* is a correct choice,

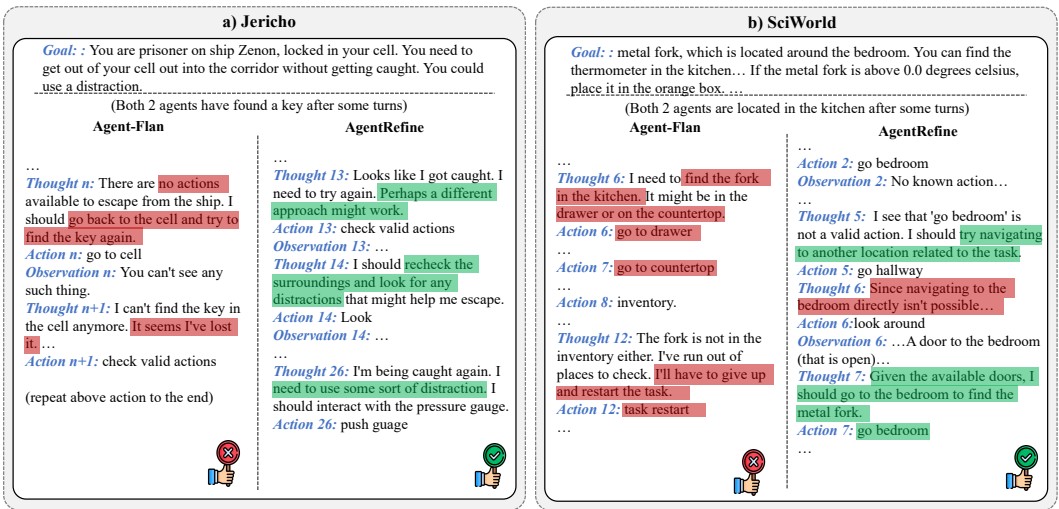

Figure 9: Comparison case study on Jericho and SciWorld between Agent-FLAN and AgentRefine.

Agent-FLAN does not correct its erroneous decision based on the returned results and repeats the *go to cell* and *check valid actions* error loop. In contrast, AgentRefine, upon realizing its actions are not achieving the goal, tries various new methods instead of endlessly repeating previously tried incorrect actions.

In Sciworld, Agent-FLAN ignores the hint in the Goal that the $fork\ is\ in\ the\ bedroom$ and chooses to search in the $kitchen$. Additionally, Agent-FLAN, having memorized the Alfworld dataset, attempts to output locations can only be found in Alfworld ($drawer$, $countertop$, and the action format $go\ to\ \{place\}$), which do not exist in SciWorld. Conversely, AgentRefine can clearly find the $thermometer$ and decide to $go\ bedroom$ to search for the $fork$. After $go\ bedroom$ fails, it decides to $go\ hallway$ based on several rounds of observation. In $Thought$ 6, although AgentRefine mistakenly believes it cannot reach the $bedroom$, its judgement shows it can revise its decisions using short-term memory (from turn 2). When $Observation$ 6 provides clear information about the $bedroom$, AgentRefine can correct its wrong decision in $Thought$ 6 and reach the $bedroom$. This indicates that AgentRefine's improvement in results is not due to memorizing prior knowledge from training data but rather its ability to efficiently utilize and integrate multiple key pieces of information from short-term memory to correct errors in historical decisions.

## 8 GPT-4 JUDGEMENT'S RELIABILITY

Figure 10 shows the comparison of GPT-4 and human judgement on whether a turn needs to be refined. We randomly sampled 50 trajectories from the generated trajectory. In each trajectory, we randomly sampled 1 right turn and 1 wrong turn. We asked the human annotator to label the correctness of the turn. The human annotator receives the historical thought, action, and observation before the right/wrong turn as well as the right/wrong turn's thought, and action in ReAct format. It also receives the script corresponding to the trajectories. The results show that in the turns that GPT-4 labeled right,

| GPT-4 / Human | Right | Wrong |
|---|---|---|
| Right | 47 | 9 |
| Wrong | 3 | 41 |

Figure 10: The comparison of GPT-4's judgement and human's judgement. The right column/line means human/GPT-4 considers this turn doesn't need to be refined. The wrong column/line means human/GPT-4 considers this turn needs to be refined.

94% are aligned with human judgment, and in the turns that GPT-4 labeled wrong, 82% are aligned with human judgment. This indicates that GPT-4's judgement is reasonable.

## 9 GENERALIZATON BETWEEN GENERAL DATA AND AGENT DATA

Both Agent-FLAN and AgentTuning have found that incorporating general data can enhance the model's generalization ability. This improvement arises from the improvement of instruction-following capability. Figure 11 shows the changes in model performance after incorporating ShareGPT. Aligned with them, we also found that general data like ShareGPT can continually improve the model's Held-out task performance.

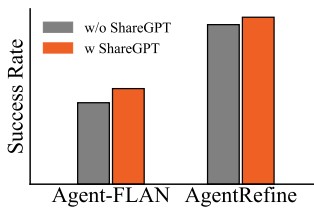

Figure 11: The success rate by incorporating ShareGPT

## 10  RELATED WORK

**Agent Finetuning** To enhance the decision-making capabilities of open-source models, a series of works currently focus on training Agent trajectories. A small number of models choose the decompose-then-execution paradigm (Yin et al., 2024), while the majority opt for using ReAct (Yao et al., 2022). Most works sample from the dataset and train the model using methods such as SFT or DPO (Rafailov et al., 2024) to improve their ability to handle Held-in problems(Zeng et al., 2023; Hu et al., 2024; Xi et al., 2024; Chen et al., 2024). AgentTuning, Agent-FLAN, and AgentGen attempt to train generalizable agent models. AgentTuning and Agent-FLAN have found that using general data like ShareGPT can improve generalization. AgentGym aims to enhance generalization by enabling the model to continuously learn new tasks and treating all tasks as Held-in. AgentGen is the first to attempt direct environment synthesis, improving generalization by enhancing the diversity of training data. In this work, we demonstrate that the above approaches still have limitations in terms of generalization, specifically in terms of easily overfitting on single data sets, getting stuck in reasoning, and learning incorrect reasoning patterns. To address this issue, we increased the diversity of training agent data through synthetic data, significantly alleviating the model's overfitting problem. Additionally, we add refinement steps in the trajectory. We show that whether the training data includes the refinement process affects the model's reasoning pattern, and adding synthetic refinement processes greatly enhances the generalization performance of LLMs.

**Data Synthesis** Due to the impending depletion of web data, the use of synthetic data has become a research hotspot. The synthesis can be divided into query synthesis and response synthesis. Most agent-tuning approaches synthesize the response in different ways like the plan (Yin et al., 2024), ReAct format (Zeng et al., 2023), JSON format (Zhang et al., 2024), chat format (Chen et al., 2024), pair format (Xiong et al., 2024), or evaluation of the state knowledge (Qiao et al., 2024), etc. The other way is to synthesize queries, like evolving a given query (Xu et al., 2023) or using pre-train data as a seed to generate new data (Chan et al., 2024). Among agent research, only AgentGen explores query synthesis. AgentRefine tries to synthesize queries and responses at the same time and uses a verifier to supervise the quality of the responses.

**Self-Refine** Self-refine refers to the process where a model iteratively generates better results through feedback. SELF-REFINE (Madaan et al., 2024; Huang et al., 2023) finds GPT-4 can find and correct mistakes itself in a compulsory pipeline - generate answer, asking a refinement advise and use the question and the advise to generate answer again. AgentRefine trains models to develop step-level refinement abilities. This means the model can spontaneously adjust its decision processes based on feedback from the environment, rather than relying on compulsory guidance from a pipeline at instance-level. AgentRefine is also the first approach to identify the connection between step-level refinement and agent generalization.

## 11  CONCLUSION

In this work, we study the generalized agent abilities for open-source LLMs via agent tuning. Current work performs well on held-in evaluation sets but fails to generalize to held-out sets because of overfitting to several manual agent environments. We present the AgentRefine approach to enable the model to correct its mistakes based on the environment feedback. Experiments demonstrate that AgentRefine significantly outperforms state-of-the-art agent-tuning work in terms of generalization ability on diverse agent benchmarks. Our analysis shows that self-refinement enables the robustness of agent capability and the diversity of agent environments and thoughts further enhances the performance. We hope to provide new insight for future agent research.

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

## ACKNOWLEDGMENT

This work was partially supported by the State Key Laboratory of Massive Personalized Customization System and Technology (No. H&C-MPC-2023-02-07(Q)), State Grid Technology Project (5700-202416236A-1-1-ZN) "Research on active semantic discovery technology based on SG-CIM and its application in power grid equipment supply chain optimization", China Unicom Software Research Institute "Framework Agreement for Seven Model Technology Research and Application Demonstration Projects (Software Development for Government Enterprise Content Generation) of China Unicom Software Research Institute from 2024 to 2025" (No.5500331818), and the National Natural Science Foundation of China (NSFC No.62076031 and No.62076036).

## ETHICS STATEMENT

When using a large amount of open-source resources for data synthesis, an important issue is the generation of harmful and malicious data. In our work, we use Persona-Hub, a synthesized dataset that has undergone security processing. We use it to synthesize tasks and environmental information, which pass our secondary review and are safe to use. However, our method may have potential risks

of misuse, such as enhancing LLM's capabilities in malicious agent tasks, like generating attack codes. Therefore, adhering to ethical guidelines is crucial to ensuring the responsible use of this technology.

## A    TASKS STATISTIC

Table 6 presents the number of test data and domains in the 5 tasks. These number calculates the Held-out Task score. Specifically, $Held - outTaskscore = (BabyAIscore * 112 + SciWorldscore * 90 + PDDLscore * 60 + Jerichoscore * 20)/282$

| task | Alfworld | BabyAI | SciWorld | PDDL | Jericho |
|------|----------|--------|----------|------|---------|
| #num | 134 | 112 | 90 | 60 | 20 |
| Domain | Science Experiment | Household Tasks | Robot Exploration | Strategy Games | Long Text Games |

Table 6: tasks statistic in AgentBoard. #num refers to the number of data for testing.

## B    THE HISTORY OF AGENT-TUNING

In recent years, LLM-Based Agents have become a popular paradigm. However, improving LLM performance on agent tasks during the post-training phase remains a challenging issue. Previous work typically sampled and trained in fixed environments (with Held-in data that is distributionally similar to the test data)(Xi et al., 2024), which significantly improved performance on specific tasks (test sets that are distributionally similar to the training data). However, performance drops sharply once the task changes.

AgentTuning (Zeng et al., 2023) was the first to recognize this issue by adding a portion of general alignment data to the single-agent data, alleviating the problem and demonstrating initial generalization capabilities. Agent-FLAN (Chen et al., 2024) further improved the single-agent data, enhancing the model's generalization in agent tasks.

In our work, we demonstrate that the above approaches still have significant limitations in terms of generalization, specifically in terms of easily overfitting on single data sets, getting stuck in reasoning, and learning incorrect reasoning patterns (as discussed in Figure 2, Figure 9, and Section 4.3, etc.). To address this issue, we increased the diversity of training agent data through synthetic data, significantly alleviating the model's overfitting problem. Additionally, we add refinement steps in the trajectory. We show that whether the training data includes the refinement process affects the model's reasoning pattern, and adding synthetic refinement processes greatly enhances the generalization performance of LLMs.

## C    SYNTHESIS DATA WITH PERSONA

Persona represents diverse and rich information content. Persona hub (Chan et al., 2024) contains 1,000,000,000 personas after filtering via diverse. If the filter cosine similarity is 0.5, it can still generate 1 million diverse personas. The persona hub also demonstrated that the data generated via the persona hub has similar diversity to the persona data and its scaling experience shows that data generated via the persona hub is not yet saturated at the size of 1M under math problem.

## D    TRAINING HYPER PARAMETER

For all models, the learning rate is 5e-6 with a cosine learning rate scheduler and no warm-up steps. The batch size is 64. The max length is 8192 for 7/8b models and 4096 for 70b models due to limited storage for DeepSpeed (Rasley et al., 2020) usage. Aligned with Agent-FLAN, we choose AgentRefine with 32000 data for the default training setting. Aligned with AgentGen (Hu et al., 2024), we train our model for 10 epochs and select the checkpoint with the best average results to report. We also modified the LLaMA-Factory's SFT loss to Equation 1. Other settings are aligned with LLaMA-Factory's default settings.

# E   COMPARISON AMONG AGENT DATASETS

Table 7 compares the number of trajectories, the methods to obtain environments and trajectories, the held-in tasks in the AgentBoard benchmark, and the availability of refinement steps among Agent-FLAN, AgentGym, AgentGen, and AgentRefine. AgentRefine can easily scale its data and includes refinement steps in the training set. AgentGen and our work are contemporary. Our commonality lies in synthesizing diverse environments, but we place more emphasis on enhancing refinement abilities.

| Method | Trajectory num | Environment construction | Trajectory construction | Held-in environment | Refinement step |
|--------|----------------|--------------------------|-------------------------|---------------------|-----------------|
| Agent-FLAN | 34440 | manual | sampled | Alfworld | No |
| AgentGym | 14485 | manual | sampled | Alfworld, BabyAI, SciWorld | No |
| AgentGen | 7246 | synthetic | sampled | N/A | No |
| AgentRefine | (max) 64000 | synthetic | synthetic | N/A | Yes |

Table 7: Comparison of AgentRefine with other method covers several aspects: the number of trajectories, the way to get environment, the way to get trajectory, the held-in task in AgentBoard, availability of refinement step

# F   IND FILTERING EXPERIMENTS

To remove the interference from IND data, we perform an experiment where we train model using data that excludes all IND training data. Agent-FLAN removes 672 samples out of 34440 samples, and AgentGym removes 5350 samples out of 14485 samples. The result in Table 8 shows that AgentRefine outperforms the other two methods in all tasks. This demonstrates that our method significantly improves over previous methods.

| Method | Alfworld | | BabyAI | | SciWorld | | PDDL | | Jericho | |
|--------|----------|----------|--------|----------|----------|----------|------|----------|---------|----------|
| | Success | Progress | Success | Progress | Success | Progress | Success | Progress | Success | Progress |
| LLaMA-3-8B-Instruct | 22.4 | 46.1 | 45.5 | 56.5 | 7.8 | 41.1 | 10.0 | 38.4 | 0.0 | 24.3 |
| AgentGen | 29.1 | 47.6 | 20.5 | 35.0 | - | - | 11.7 | 23.0 | - | - |
| AgentGym w/o ind data | 5.9 | 28.7 | 27.7 | 40.0 | 2.2 | 14.3 | 8.2 | 18.8 | 5.0 | 13.7 |
| Agent-FLAN w/o ind data | 1.5 | 19.7 | 32.1 | 45.0 | 2.2 | 12.1 | 6.6 | 23.6 | 0.0 | 14.5 |
| AgentRefine | 44.8 | 63.8 | 37.5 | 50.4 | 14.4 | 42.6 | 16.6 | 37.8 | 10.0 | 32.3 |

Table 8: IND Filtering Experiments

# G   REFLEXION EXPERIMENT

Table 9 presents the results with Reflexion (Shinn et al., 2024). It shows that AgentRefine outperforms other methods when adding Reflexion, especially in Alfworld, since AgentRefine isn't trained on any Alfworld data, yet it outperforms AgentGym, and Agent-FLAN, whose models are trained on Alfworld data. This indicates that AgentRefine can utilize Reflexion more effectively than other methods.

| Method | Alfworld | | BabyAI | | SciWorld | | PDDL | | Jericho | |
|--------|----------|----------|--------|----------|----------|----------|------|----------|---------|----------|
| | Success | Progress | Success | Progress | Success | Progress | Success | Progress | Success | Progress |
| LLaMA-3-8B-Instruct + Reflexion | 41.2 | 56.2 | 45.5 | 56.5 | 7.8 | 39.4 | 10.0 | 38.4 | 5.0 | 20.9 |
| AgentGym + Reflexion | 86.5 | 91.8 | 47.3 | 60.9 | 23.3 | 50.6 | 1.7 | 16.6 | 0.0 | 12.1 |
| Agent-FLAN + Reflexion | 83.1 | 89.4 | 32.1 | 42.3 | 5.5 | 13.1 | 10.0 | 24.8 | 0.0 | 9.7 |
| AgentRefine + Reflexion | 90.3 | 95.6 | 37.5 | 50.4 | 16.6 | 44.5 | 16.6 | 37.8 | 10.0 | 32.7 |

Table 9: Reflexion Experiment. The underlined text indicates that the training data is sampled in the same environment as the task and is considered as held-in evaluation

# H   STANDARD DEVIATIONS

Table 10 shows the average and standard deviation for each task. We use the results from Table 4 (decoding temperature = 1.0 with 10 sample times). AgentRefine's average performance exceeds

that of other methods by at least 2 standard deviations in most OOD tasks. This demonstrates that our method represents a significant improvement over previous methods.

| Model | Alfworld | | BabyAI | | SciWorld | | PDDL | | Jericho | |
|---|---|---|---|---|---|---|---|---|---|---|
| | Success | Progress | Success | Progress | Success | Progress | Success | Progress | Success | Progress |
| AgentGym | $64.3_{\pm3.3}$ | $78.0_{\pm3.1}$ | $48.2_{\pm3.3}$ | $64.2_{\pm2.3}$ | $25.5_{\pm4.7}$ | $55.4_{\pm3.2}$ | $4.5_{\pm1.8}$ | $16.9_{\pm3.1}$ | $0.0_{\pm0.0}$ | $15.3_{\pm1.5}$ |
| Agent-FLAN | $\underline{54.7}_{\pm3.9}$ | $\underline{71.6}_{\pm2.5}$ | $31.4_{\pm3.0}$ | $41.4_{\pm3.1}$ | $1.2_{\pm1.0}$ | $11.1_{\pm1.2}$ | $3.8_{\pm1.6}$ | $16.4_{\pm2.7}$ | $0.0_{\pm0.0}$ | $10.5_{\pm1.9}$ |
| AgentRefine | $\underline{60.1}_{\pm2.6}$ | $\underline{72.9}_{\pm2.4}$ | $37.6_{\pm1.3}$ | $52.2_{\pm1.9}$ | $10.4_{\pm3.2}$ | $35.0_{\pm3.2}$ | $13.2_{\pm2.0}$ | $37.4_{\pm2.2}$ | $11.0_{\pm4.6}$ | $30.9_{\pm3.2}$ |

Table 10: Model's average performance and standard deviations on different data. We used a high temperature and randomly sampled 10 times. The underlined text indicates that the training data is sampled in the same environment as the task and is considered as the held-in evaluation.

## I  ROBUSTNESS ANALYSIS WITH DIFFERENT COMPONENTS

| Model | Alfworld | | Perturbation 1 | | Perturbation 2 | | Perturbation 3 | | Perturbation 4 | | Average | | STD | |
|---|---|---|---|---|---|---|---|---|---|---|---|---|---|---|
| | Success | Progress | Success | Progress | Success | Progress | Success | Progress | Success | Progress | Success | Progress | Success | Progress |
| AgentRefine | 48.5 | 61.5 | 56.7 | 67.7 | 51.5 | 63.1 | 40.2 | 65.1 | 45.5 | 60.6 | 48.48 | 63.60 | 5.78 | 2.71 |
| - w half training data | 36.6 | 55.9 | 41.8 | 59.0 | 37.3 | 58.4 | 26.1 | 43.2 | 13.4 | 24.2 | 31.04 | 48.14 | 10.79 | 13.50 |
| - w/o refinement data | 49.3 | 65.2 | 53.7 | 69.7 | 49.2 | 65.0 | 52.9 | 65.6 | 38.8 | 59.7 | 48.78 | 65.04 | 5.47 | 3.39 |
| - w/o verification | 25.4 | 36.1 | 39.5 | 49.2 | 23.9 | 34.9 | 23.9 | 34.0 | 15.6 | 27.3 | 25.66 | 36.30 | 6.24 | 7.08 |

Table 11: Ablation study across various perturbations. We experimented with small data size (i.e.8000) and in "w half training data" setting, we use 4000 data. The w/o verification setting contains data in 3 styles: 1. The data that does not contain a refinement step. 2. The data with wrong parameter/action name but is not identified by the GPT-4. 3. The data is correct and has the refinement step (i.e. a subset of the AgentRefine data). We remove incomplete data or the data that can not be parsed into the training data

Table 11 presents the contribution to robustness among different components. When training on 4000 data, the standard deviation of the success score is almost double that of the baseline which means the number of the training data is the most important factor for the model's robustness.

## J  MODEL'S INSTRUCTION-FOLLOWING ABILITY

We use MT-bench (Zheng et al., 2023) to test models' instruction-following ability and use gpt-4o-2024-05-13 to judge the score.

The score of AgentRefine is approximately 0.2 points higher than that of Agent-FLAN regardless of whether ShareGPT is incorporated. After incorporating ShareGPT, both show an improvement of about 2 points.

| Method | MT-bench |
|---|---|
| Agent-FLAN | 3.73 |
| +ShareGPT | 5.71 |
| AgentRefine | 3.96 |
| +ShareGPT | 5.91 |

Figure 12: Model Performance on Different Tasks

## K  PERTURBATION DETAILS

We have made 5 perturbation in Alfworld:

· Perturbation 1: change $clean\ \{obj\}\ with\ \{recep\}$, $cool\ \{obj\}\ with\ \{recep\}$, $heat\ \{obj\}with\ \{recep\}$ to $clean\ \{obj\}\ using\ \{recep\}$, $cool\ \{obj\}\ using\ \{recep\}$, $heat\ \{obj\}\ using\ \{recep\}$ in the instruction

· Perturbation 2: change $go\ to\ \{recep\}$ to $move\ to\ \{recep\}$ in the instruction

· Perturbation 3: change $take\ \{obj\}\ from\ \{recep\}$ to $from\ \{recep\}\ take\ \{obj\}$ in the instruction

· Perturbation 4: delete all space between item name and item number in the instruction.

· Perturbation 5: remove all IND data in the training set and retrain the model.

We also revise the environment to adjust to these changes.

## L  SCRIPT GENERATION

```
{
    "Thought" : (string, compulsory) "The design of the
        environment, goal and available actions of the
        player to achieve.",
    "Environment" : {
        "initial state" : (string, compulsory) "The
            initial state of the environment.",
        "places and objects" : {
            "<The name of the place or object>" : {
                "information" : (string, optional) "The
                    information of the  place or object,
                    which will only be shown to player
                    when the object is examined/opened/
                    looked or the player have just step in
                     its receptacle etc.",
                "<The information of the place or object
                    >" : (string, optional) "The
                    information of the place or object,
                    which will only be provide to DM",
                "<The name of the  place or object>" : {
                    "information" : (string, optional) "
                        The information of the place or
                        object, which will only be shown
                        to player when the object is
                        examined/opened/looked or the
                        player have just step in its
                        receptacle etc. It must be
                        concrete (for example, if you add
                        information in a document, you
                        need to give the important part of
                         the document context instead of a
                         brief introduction.).",
                    "location" : (string, optional) "The
                        relative location between the
                        object/place and its json upper
                        level object/place (i.e.
                        receptacle).",
                    "relative location" : (list of string
                        , optional) ["The relative
                        location of the places or objects
                        in the same json level."]
                }
            },
            "relative location" : (list of string,
                optional) ["The relative location of the
                places or objects in the same json level
                ."]
        },
        "player":{
            "information": (string, compulsory) "The
                player's restrictions."
        }
    },
```

```
    "Goal" : (string, compulsory) "The goal of the player
        to achieve. It need to be clear(has unique and
        concrete completion conditions), achievable and
        can be finished by one person.",
    "Completion Conditions" : (list of string, compulsory
        ) [
            "The specific conditions that the player must
                meet to complete the task."
        ],
    "Available Actions" : {
        "<The name of the action>" : {
            "description" : (string, optional) "The
                description of the action.",
            "special format" : (string, optional) "The
                special format of the action. Only when
                the parameter is not in the place/object
                and their information above can use this
                key. (This key is compulsory when
                answering the question and editing the
                code.)",
            "verification code" : (string, compulsory) "
                The regular expression of the action.",
            "parameters" : {
                "<The name of the parameter>" :  (list of
                     string, optional) ["The value of the
                    parameter if action has placeholder.
                    Remember all possible parameter (the
                    possible place, possible object or the
                     possible item/text in the \"
                    information\" of place/object or the
                    imformation in the completion
                    conditions) should be in the list. DM
                    will strictly check the player's
                    actions according to the given
                    parameters. So you should give all
                    possible parameters with correct name
                    "]
            }
        }
    }
}
```

## M   TRAJECTORY GENERATION

```
Trajectory Generation Format

[
    {
        "turn": (int, compulsory) "The turn number, the
            first turn number should be 0, DM's turn
            number should be even.",
        "role": (compulsory) "DM",
```

```
            "Thought": (string, compulsory) "The thought of
                the DM, contains the analyze of the knowledge
                the player have known and the chain-of-thought
                 to decide the observation.",
            "Observation": (string, compulsory) "The
                observation of the DM, contains the
                information the player should know.",
            "parameter_error": (bool, compulsory) "The error
                log of the DM, if the player's last action did
                 not match the format of the available actions
                ",
            "place_error": (bool, compulsory) "The error log
                of the DM, if the player's last action act at
                a wrong place",
            "logic_error": (bool, compulsory) "The error log
                of the DM, if the player's last action matches
                 the available action but the observation is
                not changed under the action or went back to
                the sitiuation that history has been. (for
                example, go north then go south)",
            "progress_rate": (float, compulsory) "The
                progress rate of the task, the max value
                should be 1.0 which means task finsihed.",
            "finished": (bool, compulsory) "The flag of the
                task, if the task is finished, the value
                should be true."
        },
        {
            "turn": (int, compulsory) "The turn number, the
                first turn number should be 1, Player's turn
                number should be odd.",
            "role": (compulsory) "Player",
            "Thought": (string, compulsory) "The thought of
                the Player, contains the chain-of-thought to
                decide the action. You should remove the \"
                Thought:\" at the beginning of this string in
                the json output, although DM should ask for
                this format in the first turn.",
            "Action": (string, compulsory) "The action of the
                 Player, its format and the parameter MUST
                follow the script. You should remove the \"
                Action:\" at the beginning of this string in
                the json output, although DM should ask for
                this format in the first turn."
        }
    ]
```

## N  ERROR TURN STATISTICS

Figure 13 presents the error turn statistics in AgentRefine (32000). Most of the error-refine pairs consist of one turn, which accounts for about 16% among all turns. However, AgentRefine also includes error-refine pairs whose lengths exceed three turns.

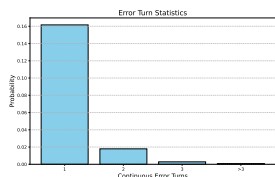

Figure 13: The statistics of Continuous Error Turns in AgentRefine

## O  TRAJECTORY VERIFICATION

Algorithm 1 presents the Trajectory Verification pipeline.

---

**Algorithm 1** Trajectory Verification

---

1: Input: Available Actions, Trajectory, Verified Trajectory
2: # The Verified Trajectory will be set to an empty list if this is the first verification of the persona or the last generation's fault is $error\_num \leq 1$
3: Initialize: error_num=0
4: **if** JSON format verification does not pass **then**
5:     JSON format verification does not pass
6: **end if**
7: **for** turn in Trajectory **do**
8:     **if** JSON keys in turn do not match the requirement **then**
9:         return Verified Trajectory and the signal
10:     **end if**
11:     **if** Player's turn **then**
12:         # We only check the action when DM considers it correct.
13:         **if** $not$ next DM turn shows error signal **then**
14:             **if** Player's action doesn't match any $action_i$ (and its parameter) in Available Actions **then**
15:                 return Verified Trajectory and the signal
16:             **end if**
17:         **end if**
18:     **end if**
19:     **if** DM's turn **then**
20:         **if** Error signal **then**
21:             error_num += 1
22:         **end if**
23:         **if** This is the last turn **then**
24:             # The last turn should not have any error
25:             **if** Error signal **then**
26:                 return Verified Trajectory and the signal
27:             **end if**
28:             # The last turn should finish the task
29:             **if** No 'Task Succeed' in Observation **then**
30:                 return Verified Trajectory and the signal
31:             **end if**
32:             # We need at least 2 error-refine turns.
33:             **if** $error\_num \leq 1$ **then**
34:                 return Verified Trajectory and the signal
35:             **end if**
36:         **end if**
37:     **end if**
38:     Verified Trajectory ← Verified Trajectory + turn
39: **end for**

---

