# OpenReview forum: "AgentRefine: Enhancing Agent Generalization through Refinement Tuning"
_ICLR.cc/2025/Conference — ICLR 2025 Poster_

### Official Review · Reviewer_opnM · 2024-11-03

**Soundness:** 3
**Presentation:** 2
**Contribution:** 2
**Rating:** 5
**Confidence:** 3

**Summary:**

The paper proposes AgentRefine, a framework designed to enhance the generalization capabilities of large language model (LLM)-based agents through a self-refinement process. The core idea is to enable agents to learn from their mistakes by refining their actions based on feedback from the environment. The authors introduce a data generation pipeline that simulates diverse environments and tasks, followed by a refinement tuning process to improve agent robustness and generalization. Experimental results show that AgentRefine outperforms state-of-the-art methods in held-out tasks, demonstrating improved generalization and robustness.

**Strengths:**

1. The introduction of a self-refinement process for agent tuning is a novel contribution to the field. By allowing agents to correct their mistakes based on environmental feedback, the authors propose an interesting alternative to traditional fine-tuning methods.
2. The use of diverse environments and tasks in data generation helps mitigate overfitting to specific scenarios, which is a common issue in LLM-based agents.
3. The experiments show that AgentRefine outperforms baselines in held-out tasks, suggesting that the approach has potential for improving generalization.

**Weaknesses:**

1.  The paper relies heavily on GPT-4 for generating both scripts and trajectories. This raises several concerns:
   - The quality of the generated data depends entirely on GPT-4's ability to detect and correct errors
   - The method is not truly "self-refinement" since it requires external stronger models for error detection and correction
   - The authors should analyze what happens when using weaker LLMs for data generation and verification

2. The verification process has potential flaws:
  - It uses LLMs to verify the correctness of scripts and trajectories without human validation
  - The paper lacks analysis of verification failure cases or error rates
  - The authors should include human evaluation of the verification process accuracy

3. While the paper shows improved performance, it lacks analysis of whether this is simply distillation from GPT-4 rather than true generalization and how much of the improvement comes from the refinement process versus having access to GPT-4's knowledge

4. The experiments only scale up to 64k examples. Would the computational cost of generating refinement data with GPT-4 makes large-scale training difficult? Also, the authors should analyze the cost-benefit tradeoff of generating more refinement data

5. While the paper shows some robustness analysis, the perturbation experiments are limited to only action descriptions. More diverse types of perturbations should be tested. The analysis should include how different components (script generation, verification, refinement) contribute to robustness

**Questions:**

See above weakness section

---

> ### Author Response · Authors · 2024-11-17
> **Official Comment to Reviewer opnM (1/3)**
>
> Thanks for the reviewer's comments. Here are our responses to the comments.
>
> ---
>
> **Weakness 1**: The paper relies heavily on GPT-4 for generating both scripts and trajectories. This raises several concerns:
> - The quality of the generated data depends entirely on GPT-4's ability to detect and correct errors
> - The method is not truly "self-refinement" since it requires stronger external models for error detection and correction
> - The authors should analyze what happens when using weaker LLMs for data generation and verification
>
> **Response to weakness 1(1/3)**:
>
> Apologies for the confusion. We need to clarify that we have the **rule-based verification process** to detect the parameter errors, the specific process is in Appendix O, Algorithm 1. To further prove the reliability of GPT-4's judgement, we conducted an experiment to evaluate the quality of the generated data. The results below prove that **GPT-4's judgement is reliable** since 88% (47+41) of the judgement is consistent with human annotators. We add this experiment in Section 8. Thanks for the reviewer!
>
> | | Right turn in GPT-4's  judgement | Wrong turn in GPT-4's  judgement |
> |---|---|---|
> | Right turn in  human annotator's judgement| 47 | 9 |
> | Wrong turn in  human annotator's judgement| 3 | 41 |
>
> Settings: We randomly sampled 50 trajectories from the generated trajectory. In each trajectory, we randomly sampled 1 right turn and 1 wrong turn. We asked the human annotator to label the correctness of the turn. The human annotator can receive the historical thought, action, and observation before the right/wrong turn and right/wrong turn's thought, and action in ReAct format.
>
>
> **Response to weakness 1(2/3)**:
>
>  Apologies for the confusion. We need to clarify that we **only use GPT-4 to generate the training trajectories**, and we **do not use GPT-4 to detect an error in evaluation**. In evaluation, the AgentRefine model should be able to detect errors, correct errors, and think of multiple paths when it encounters a mistake by itself. We believe this is a form of self-refinement.
>
> **Response to weakness 1(3/3)**:
>
>  Thanks for the suggestion. We conducted an experiment using **opensource model** Deepseek-v2.5 to generate environments and trajectories in Section 5 (Appendix F in the original paper). Deepseek-v2.5 is weaker than GPT-4. The results show that the performance of **the model trained with data from the open-source model is still better than the model trained with Agent-Flan** (whose data comes from GPT-4).
>
> | Method            | Alfworld       |            | BabyAI        |            | SciWorld      |            | PDDL          |            | Jericho       |            |
> |-------------------|----------------|------------|---------------|------------|---------------|------------|---------------|------------|---------------|------------|
> |                   | Success        | Progress   | Success       | Progress   | Success       | Progress   | Success       | Progress   | Success       | Progress   |
> | Agent-FLAN          | _76.2_             | _79.7_     | 25.0  | 35.3     | 1.1    | 10.9     | 8.3                    | 25.5                   | 0.0                  | 10.1                 |
> | AgentRefine-DeepSeek| 32.0                         | 44.2                       | 36.6                        | 48.1                     | 2.2                       | 21.6                     | 16.6                   | 36.7                   | 5.0                  | 29.0                 |
> | AgentRefine-GPT-4o  | 36.6                         | 55.9                       | 33.9                        | 44.1                     | 11.1                      | 31.4                     | 18.3                   | 37.9                   | 10.0                 | 28.8                 |
>
> The underlined text indicates that the training data is sampled in the same environment as the task and is considered as held-in evaluation. We use 4000 data to train AgentRefine-DeepSeek and  AgentRefine-GPT-4o.
>
> ---
>
> **Weakness 2**: The verification process has potential flaws:
> - It uses LLMs to verify the correctness of scripts and trajectories without human validation
> - The paper lacks analysis of verification failure cases or error rates
> - The authors should include human evaluation of the verification process accuracy
>
> **Response to weakness 2 (1/2)**:
>
> Apologies for the confusion. We need to clarify that our verification process is rule-based. The GPT-4's judgement is only used when generating the trajectory. To clarify the trajectory generation process, we update Figure 4 in the paper. Thanks for the suggestion.
>
> **Response to weakness 2 (2/2)**:
>
>  Thanks for the suggestion. We conducted an experiment to evaluate the quality of the generated data in the table above. The results prove that GPT-4's judgement is reliable since 88% (47+41) of the judgement are consistent with human annotators. We will include this analysis in the final version of the paper.

---

> ### Author Response · Authors · 2024-11-17
> **Official Comment to Reviewer opnM (2/3)**
>
> **Weakness 3**: While the paper shows improved performance, it lacks analysis of whether this is simply distillation from GPT-4 rather than true generalization and how much of the improvement comes from the refinement process versus having access to GPT-4's knowledge
>
> **Response to weakness 3**:
>
> Apologies for the confusion. We conducted an experiment in Section 4.2 Table 2 to analyze the improvement from the refinement process. All models are trained with data generated by GPT-4. The result proved that AgentRefine model outperforms other models, so **the improvement does not come from distillation from GPT-4/access to GPT-4's knowledge**. The AgentRefine model is better than the model trained with the trajectory that does not contain refinement step (w/o refinement data) or the model trained with trajectory that contains the refinement step but masks the refinement step loss (w/o refinement loss). The result shows that the **refinement process is important for the model to generalize well**.
>
> | Method                 | Alfworld |          | BabyAI  |          | SciWorld |          | PDDL    |          | Jericho |          |
> |------------------------|----------|----------|---------|----------|----------|----------|---------|----------|---------|----------|
> |                        | Success  | Progress | Success | Progress | Success  | Progress | Success | Progress | Success | Progress |
> | AgentRefine            | 48.5     | 61.5     | 37.1    | 51.7     | 7.7      | 33.1     | 21.7    | 37.4     | 5.0     | 26.2     |
> | - w/o refinement loss  | 40.3     | 58.8     | 34.8    | 45.6     | 4.4      | 22.7     | 20.0    | 37.4     | 0.0     | 16.1     |
> | - w/o refinement data  | 49.3     | 65.2     | 30.4    | 43.1     | 5.5      | 21.3     | 11.7    | 32.5     | 0.0     | 13.8     |
> | - w erroneous loss     | 29.9     | 43.9     | 23.2    | 31.6     | 3.3      | 19.0     | 8.3     | 28.3     | 5.0     | 18.4     |
>
> ---
>
> **Weakness 4**: The experiments only scale up to 64k examples. Would the computational cost of generating refinement data with GPT-4 make large-scale training difficult? Also, the authors should analyze the cost-benefit tradeoff of generating more refinement data
>
> **Response to weakness 4 (1/3)**:
>
>  Thanks for the suggestion. We need to clarify that **generating a single trajectory does not rely on other trajectories**. The diversity of the trajectory is guaranteed by the diversity of the persona (Page 5, last sentence in the second paragraph of section Embedding-based Deduplication in Persona Hub[1]). Since the persona is diverse in almost 1M data (Figure 9 in Persona Hub), the tasks are diverse as well (Figure 10 in Persona Hub). So, **the cost is linear to the number of trajectories**. This is not a problem for large-scale training.
>
> **Response to weakness 4 (2/3)**:
>
> Thanks for the suggestion. Figure 5's result shows that the model's performance is almost linear to the log of the number of trajectories. So the model's performance is almost the log of the cost. The cost-benefit tradeoff point is based on the specific entity's preference so it's hard to calculate without knowing the preference, but the user(entity) can calculate it based on the log curve of cost and performance.
>
> **Response to weakness 4 (3/3)**:
>
> Because of the source we have, we generated 64k training data.  As a result, there are 5 different sizes (4k, 8k, 16k, 32k, 64k) in our scaling experiment which is widely used in other papers[2].

---

> ### Author Response · Authors · 2024-11-17
> **Official Comment to Reviewer opnM (3/3)**
>
> **Weakness 5**: While the paper shows some robustness analysis, the perturbation experiments are limited to only action descriptions. More diverse types of perturbations should be tested. The analysis should include how different components (script generation, verification, refinement) contribute to robustness
>
> **Response to weakness 5 (1/2)**:
>
> Thanks for the suggestion. We conducted an experiment to test the performance of the model trained with different types of perturbations in the table below, the result shows that **AgentRefine is more robust than other methods**. We revise Table 3 and add the introduction of Perturbation 4 and Perturbation 5 in Appendix K. Thanks for the reviewer!
>
> | Model            | Alfworld      |              | P 1 |              | P 2 |              | P 3 |              | P 4 |              | P 5 |        | Average |      | STD | |
> |------------------|---------------|--------------|----------------|--------------|----------------|--------------|----------------|--------------|----------------|--------------|----------------|--------------|----------------------|--------------|------------------|--------------|
> |    | Success| Progress| Success| Progress| Success| Progress| Success| Progress| Success | Progress     | Success| Progress|Success| Progress|Success| Progress|
> | LLaMA3-8B-Instruct   | 22.4 | 46.1   | 23.1| 45.6| 24.6 | 45.0 | 17.9  | 45.1 | 17.9  | 45.1 | 22.4| 46.1  | 21.4| 45.5  |2.68|0.47|
> | AgentGym         | 61.9     | 76.9     | 29.1   | 59.2   | 49.2   | 65.3     | 32.8 | 53.9  | 38.8     | 48.2  | 5.9   | 28.7  | 36.3 | 55.4 |19.97|16.66|
> | Agent-Flan   | 67.2| 79.7| 21.6| 58.8| 51.4 | 71.3 | 27.6 | 53.5 | 52.2 | 67.9  | 1.5| 19.7  | 36.9  | 58.5|21.98|22.53|
> | AgentRefine      | 44.8 | 63.8| 50.0  | 66.5 | 51.5| 66.7| 54.5  | 70.0  | 45.5  | 60.6  | 44.8 | 63.8 | 48.5  | 65.2 |3.73|3.56|
>
> Note 1:  P denotes Perturbation
>
> Note 2: The number of AgentRefine training data is 32000.
>
> Note 3:  Perturbation 1-3's setting is the same as the setting in the paper. Perturbation 4 is to change the item name (remove the space between the item and its number) in the prompt (for example sofa 1 -> sofa1). Perturbation 5 filters out the IND data in the training data and retrains the model. Since AgentRefine and  LLaMA3-8B-Instruct do not use the IND data, the performance of the model is not changed. The result shows that the model trained with AgentRefine is more robust than agent-finetuning models.
>
> **Response to weakness 5 (2/2)**:
>
>  Thanks for the suggestion. We conducted an experiment to analyze how different components contribute to robustness, as shown in the table below. The result shows that **the number of the training data** is the most important factor for the model's robustness. The **verification process** is the second important factor. We include this analysis in Appendix I. Thanks for the reviewer!
>
> | Model            | Alfworld      |              | P 1 |              | P 2 |              | P 3 |              | P 4 |              | Average |              | STD |              |
> |------------------|---------------|--------------|----------------|--------------|----------------|--------------|----------------|--------------|----------------|--------------|----------------|--------------|----------------------|--------------|
> |                  | Success       | Progress     | Success        | Progress     | Success        | Progress     | Success        | Progress     | Success        | Progress     | Success        | Progress     | Success    | Progress     |
> | AgentRefine | 48.5 | 61.5 | 56.7 | 67.7 | 51.5 | 63.1 | 40.2 | 65.1 | 45.5 | 60.6 | 48.48 | 63.60 | 5.78 | 2.71 |
> | AgentRefine(4000) | 36.6 | 55.9 | 41.8 | 59.0 | 37.3 | 58.4 | 26.1 | 43.2 | 13.4 | 24.2 | 31.04 | 48.14 | 10.79 | 13.50 |
> | AgentRefine(w/o Refinement data) | 49.3 | 65.2 | 53.7 | 69.7 | 49.2 | 65.0 | 52.9 | 65.6 | 38.8 | 59.7 | 48.78 | 65.04 | 5.47 | 3.39 |
> | AgentRefine(w/o verification) | 25.4 | 36.1 | 39.5 | 49.2 | 23.9 | 34.9 | 23.9 | 34.0 | 15.6 | 27.3 | 25.66 | 36.30 | 6.24 | 7.08 |
>
> Note 1:  P denotes Perturbation
>
> Note 2: Except for the AgentRefine(4000) setting, the number of training data is 8000.
>
> Note 3: Perturbation 1-3's setting is the same as the setting in the paper. Perturbation 4 is to change the item name (remove the space between the item and its number) in the prompt (for example sofa 1 -> sofa1).
>
> Note 4: The AgentRefine(w/o verification) setting contains data in 3 styles: 1. The data that does not contain a refinement step. 2. The data has the wrong parameter/action name but the GPT-4 does not find it. 3. The data is correct and has the refinement step (i.e. a subset of the AgentRefine data). We remove the incomplete data or the data that can not be parsed into the training data.
>
>
>
> [1] Scaling Synthetic Data Creation with 1,000,000,000 Personas
>
> [2] How Abilities in Large Language Models are Affected by Supervised Fine-tuning Data Composition

---

> ### Author Response · Authors · 2024-12-03
> **A Kind Reminder for Reading the Response**
>
> Dear Reviewer opnM,
>
> Thank you for your insightful suggestions. We have revised the paper:
> 1. We added the GPT-4 judgment reliability experiment (weakness 1-1, weakness 2)
> 2. We added the robustness components experiments and 2 more perturbation experiments (weakness 5).
> 3. We adjusted the structure of the paper and moved the open-source model experiment into the main text. (Weakness 1-3)
> 4. We explained Weakness 1-2,  Weakness 2-1, Weakness 3, and Weakness 4 in the response above, using the data in the paper.
>
> Since the rebuttal period is closing very soon, could you please check our response to see whether it mitigates your concerns? We would greatly appreciate that!
>
> If you have any further questions, please feel free to ask.
>
> Thank you for your time and consideration,
>
> The ICLR 2025 Conference Submission14212 Authors

---

### Official Review · Reviewer_NknQ · 2024-11-06

**Soundness:** 3
**Presentation:** 2
**Contribution:** 3
**Rating:** 6
**Confidence:** 3

**Summary:**

This paper discusses using synthetic data to improve the generalization ability of agents on held-out sets. Previous agent-tuning work often chose to construct agent-tuning data on held-in sets. The authors demonstrate that although these methods can greatly improve the performance of agents on held-in sets, they usually lead to overfitting, which in turn affects the performance of agents on held-out sets. Based on this observation, the authors propose AgentRefine. This method does not use task-related information at all. Instead, it uses LLM to complete the entire data generation process, including task generation, trajectory generation, and verification to construct the agent-tuning dataset, thus avoiding the possibility of overfitting to held-in sets from the very start. In the constructed dataset, the authors emphasize the ability of the agent to correct errors based on the feedback, which further improves the agent's generalization ability. They validate AgentRefine in multiple scenarios, and the experimental results show that finetuned agents outperform other baselines on held-out sets.

**Strengths:**

1. This paper discusses the generalization ability of agents, which is a very important topic for the community.

2. The authors provide quantitative analysis to explain their insight, which is very convincing.

3. Synthesizing data with almost no task-specific information is a very practical setting, and the improvement of generalization ability in this paper is impressive.

**Weaknesses:**

1. The presentation of this paper should be improved and some grammar mistakes should be fixed.

2. Some important baselines, for example, Reflexion[1], are missing and should be included.

3. They only consider decision-making tasks in their experiments. However, as they claimed on the generalization ability, tasks of different types should also be included, for example, reasoning tasks.

[1] Shinn, Noah, et al. "Reflexion: Language agents with verbal reinforcement learning." NeurIPS, 2023.

**Questions:**

1. Can you also provide more detailed statistics of your experiments, for example, the std of each task?

2. How does the agent get an error signal during the evaluation?

3. For the LLaMA-3-70B Series, the performance of AgentRefine is worse than the base model? Am I misunderstanding something?

---

> ### Author Response · Authors · 2024-11-17
> **Official Comment to Reviewer NknQ (1/2)**
>
> Thanks for the reviewer's comments. Here are our responses to the comments.
>
> ---
>
> **Weakness 1**:  The presentation of this paper should be improved and some grammar mistakes should be fixed.
>
> **Response to weakness 1**:
>
>  Thanks for the suggestion. We have checked the grammar and presentation mistakes in the paper, and have corrected them in the new version.
>
> ---
>
> **Weakness 2**: Some important baselines, for example, Reflexion[1], are missing and should be included.
>
> **Response to weakness 2**:
>
> Thanks for the reviewer's suggestion. We need to clarify that Reflexion is a method to use long-term memory instead of agent-tuning methods so we did not choose it as the baseline in the initial version. To further prove the effectiveness of AgentRefine, we add the comparison with the Reflexion+agent-tuning setting in the table below, the result proves that **AgentRefine is better than other methods in the Reflexion+agent-tuning setting.** We include this analysis in Appendix G. Thanks for the reviewer's suggestion!
>
> | Method            | Alfworld |            | BabyAI  |            | SciWorld |            | PDDL    |            | Jericho |            |
> |-------------------|----------|------------|---------|------------|----------|------------|---------|------------|---------|------------|
> |                   | Success  | Progress   | Success | Progress   | Success  | Progress   | Success | Progress   | Success | Progress   |
> | LLaMA-3-8B-chat + Reflexion   | 41.2     | 56.2       | 45.5    | 56.5       | 7.8      | 39.4       | 10.0      | 38.4       | 5.0      | 20.9       |
> | AgentGym  + Reflexion        | _86.5_    | _91.8_       | _47.3_    | _60.9_       | _23.3_   | _50.6_      | 1.7     | 16.6       | 0.0      | 12.1       |
> | Agent-Flan  + Reflexion       | _83.1_     | _89.4_      | 32.1    | 42.3       | 5.5      | 13.1       | 10.0      | 24.8       | 0.0       | 9.7        |
> | AgentRefine  + Reflexion     | 90.3     | 95.6       | 37.5    | 50.4       | 16.6     | 44.5       | 16.6    | 37.8       | 10.0      | 32.7       |
>
> The italic text indicates that the training data is sampled in the same environment as the task and is considered as held-in evaluation. AgentRefine setting is the same as the setting in the main result.
>
> ---
>
> **Weakness 3**: They only consider decision-making tasks in their experiments. However, as they claimed on the generalization ability, tasks of different types should also be included, for example, reasoning tasks.
>
> **Response to weakness 3**:
>
> Thanks for the suggestion. We add the HotpotQA [2] experiment below. HotpotQA is the reasoning task used in ReAct [3]. The result shows that **AgentRefine outperforms other methods in the reasoning task.** We include this analysis in Section 6. Thanks for the reviewer's suggestion!
>
> | Method            | HotpotQA       |            |
> |-------------------|----------------|------------|
> |                   | EM        | F1   |
> | LLaMA-3-8B-Instruct  | 29.3     | 36.6      |
> | AgentGym          | 28.0          | 37.4      |
> | Agent-FLAN        | 24.6          | 32.4      |
> | AgentRefine       | 37.0         | 44.6      |
>
> We use Wikipedia search in LATS [4] as the environment. We randomly sample 300 questions from HotpotQA and test the exact match (EM) and F1 score of those methods.

---

> ### Author Response · Authors · 2024-11-17
> **Official Comment to Reviewer NknQ (2/2)**
>
> **Question 1**: Can you also provide more detailed statistics of your experiments, for example, the std of each task?
>
> **Response to question 1**:
>
> Thanks for the suggestion. The table below shows the average and standard deviation of each task.  We conduct this experiment under decoding temperature = 1.0 and 10 seeds. (which is the same as the setting in BON, Table 4). **AgentRefine's average performance is greater than other methods in at least 2 standard deviations in most OOD tasks**. This demonstrates that our method is a strong improvement over previous methods. We include this analysis in Appendix H. Thanks for the reviewer!
>
> | Method            | Alfworld      |            | BabyAI        |            | SciWorld     |            | PDDL         |            | Jericho      |            |
> |-------------------|---------------|------------|---------------|------------|--------------|------------|--------------|------------|--------------|------------|
> |                   | Success       | Progress   | Success       | Progress   | Success      | Progress   | Success      | Progress   | Success      | Progress   |
> | AgentGym          |  _64.3 (3.3)_    |  _78.0 (3.1)_  | _48.2 (3.3)_     | _64.2 (2.3)_| _25.5 (4.7)_  | _55.4 (3.2)_ | 4.5 (1.8)    | 16.9 (3.1) | 0.0 (0.0)    | 15.3 (1.5) |
> | Agent-Flan        | _54.7 (3.9)_   | _71.6 (2.5)_| 31.4 (3.0)    | 41.4 (3.1) | 1.2 (1.0)    | 11.1 (1.2) | 3.8 (1.6)    | 16.4 (2.7) | 0.0 (0.0)    | 10.5 (1.9) |
> | AgentRefine       | 60.1 (2.6)    | 72.9 (2.4) | 37.6 (1.3)    | 52.2 (1.9) | 10.4 (3.2)   | 35.0 (3.2) | 13.2 (2.0)   | 37.4 (2.2) | 11.0 (4.6)   | 30.9 (3.2) |
>
>
> The italic text indicates that the training data is sampled in the same environment as the task and is considered as the held-in evaluation.  The data format is average (std).
>
> ---
>
> **Question 2**: How does the agent get an error signal during the evaluation?
>
> **Response to question 2**:
>
> Apologies for the confusion. The agent only gets the signal from the environment. The agent should be able to detect errors, correct errors, and think of multiple paths based on the environmental signal by itself. We **do not use GPT-4 in evaluation**.  We believe this is a form of self-refinement.
>
> ---
>
> **Question 3**: For the LLaMA-3-70B Series, the performance of AgentRefine is worse than the base model?
>
> **Response to question 3**:
>
>  Apologies for the confusion. We need to emphasize that we use the **base model (LLaMA-3-70B-Base)** to train the AgentRefine, AgentGym, and Agent-Flan, to **get rid of the influence of the post-training**. The LLaMA-3-70B model to be evaluated is the **LLaMA-3-70B-Instruct** model which is already trained with more than 10 million SFT data and RLHF (which is close source). So, comparing the performance of AgentRefine with AgentGym and Agent-Flan is more reasonable.
>
>
> [1] Reflexion: Language agents with verbal reinforcement learning.
>
> [2] Hotpotqa: A dataset for diverse, explainable multi-hop question answering.
>
> [3] React: Synergizing reasoning and acting in language models
>
> [4] Language Agent Tree Search Unifies Reasoning Acting and Planning in Language Models

---

> > ### Comment · Reviewer_NknQ · 2024-11-21
> >
> > Thank you for your detailed response. I appreciate the clarifications and additional experiments provided in the authors' rebuttal, which have addressed most of my concerns. I have raised my score accordingly.

---

> ### Author Response · Authors · 2024-12-01
> **Thank you for Reviewer's reply**
>
> Dear Reviewer NknQ,
>
> We are pleased to see that our response has alleviated your concerns. Your suggestions have all been very helpful to us.Thank you again for your thorough review and suggestions regarding our work.
>
> ICLR 2025 Submission14212 Author

---

### Official Review · Reviewer_HUkR · 2024-11-07

**Soundness:** 2
**Presentation:** 3
**Contribution:** 2
**Rating:** 6
**Confidence:** 3

**Summary:**

The paper presents a framework aimed at improving the generalization capabilities of Large Language Model (LLM) based agents through instruction tuning. The authors observe that existing agent training methods overfit to specific environments and struggle with new situations, leading to poor generalization. To address this, they propose AgentRefine, which incorporates self-refinement processes to enable the model to learn from its mistakes and adapt to diverse environments and tasks.

**Strengths:**

1. The paper is well-organized and easy to follow, with a clear progression from motivation to methodology.
2. The identification of the generalization gap in existing LLM-based agents and the proposal of a self-refinement approach to address it is a rational step forward in the field.

**Weaknesses:**

1. The problem of generalization in LLM-based agents has been extensively discussed in previous literature, making the contribution of this work less novel. For example,  [1]  investigates the robustness of accuracy measurements in large language models (LLMs) when the order of answer labels is shuffled, using the MMLU dataset as a testbed.
2. The methodology, while intuitive, lacks significant innovation, as the approach of enhancing generalization through data synthesis is not new [2].
3. The experimental results do not demonstrate a strong improvement over existing methods, which questions the practical impact of the proposed approach. An apple-to-apples comparison of your main results to show the advantage of the algorithm would make your results more straightforward and strong, instead of using a lot of underlined text to filter out the results where training data is sampled in the same environment as the task.

[1] Changing Answer Order Can Decrease MMLU Accuracy.

[2] Knowledgeable Agents by Offline Reinforcement Learning from Large Language Model Rollouts.

**Questions:**

see weaknesses

---

> ### Author Response · Authors · 2024-11-17
> **Official Comment to Reviewer HUkR (1/2)**
>
> Thanks for reviewer's comments. Here are our responses to the comments.
>
> ---
>
> **Weakness 1**:  The problem of generalization in LLM-based agents has been extensively discussed in previous literature, making the contribution of this work less novel. For example, [1] investigates the robustness of accuracy measurements in large language models (LLMs) when the order of answer labels is shuffled, using the MMLU dataset as a testbed.
>
>
> **Response to weakness 1 (1/2)**:
>
> Apologies for the confusion. We appreciate the opportunity to clarify the differences between our work and previous studies.
>
> In recent years, LLM-Based Agents have become a popular paradigm. However, improving LLM performance on agent tasks during the post-training phase remains a challenging issue. Previous work typically sampled and trained in fixed environments (with Held-in data that is distributionally similar to the test data)\citep{xi2024AgentGym}, which significantly improved performance on specific tasks (test sets that are distributionally similar to the training data). However, performance drops sharply once the task changes.
>
> AgentTuning was the first to recognize this issue by adding a portion of general alignment data to the single-agent data, alleviating the problem and demonstrating initial generalization capabilities. Agent-FLAN further improved the single-agent data, enhancing the model's generalization in agent tasks.
>
> In our work, we demonstrate that the above approaches still have significant limitations in terms of generalization, specifically in terms of easily overfitting on single data sets, getting stuck in reasoning, and learning incorrect reasoning patterns (as discussed in Figure 2, Figure 9 and Section 4.3, etc.). To address this issue, we increased the diversity of training agent data through synthetic data, significantly alleviating the model's overfitting problem. Additionally, we add refinement steps in the trajectory. We show that whether the training data includes the refinement process affects the model's reasoning pattern, and adding synthetic refinement processes greatly enhances the generalization performance of LLMs.
>
>
>
> ---
>
> **Response to weakness 1 (2/2)**:
>
> Thanks for your valuable feedback. In our experiments, we did indeed use methods similar to [1]: adding perturbations to observe changes in model performance. However, we would like to clarify that, although our experimental methods are similar, **the conclusions and findings are entirely different**. By adding perturbations in Alfworld, we found that previous work resulted in significant performance degradation because these works used Held-in training data that is distributionally similar to Alfworld. We demonstrate that this **performance drop is a form of overfitting**, where the model overfits to simply memorizing actions rather than truly learning a suitable meta-algorithm for agent tasks. **Because of this, we create AgentRefine**, which does not use any IND data.
>
> In contrast, [1] merely observed a lack of robustness without an in-depth explanation. Moreover, **their experimental setup did not involve training with Held-in data**, and **the performance degradation was** not due to overfitting, but more likely due to model preferences for option positions and pre-training data leakage, **among other reasons**.
>
> Therefore, we believe that although our work and previous work used similar methods, the conclusions drawn and the improvements to the methods are significantly different. We will reorganize and clarify these differences in our paper to help readers better understand our work.
>
> ---
>
> **Weakness 2**: The methodology, while intuitive, lacks significant innovation, as the approach of enhancing generalization through data synthesis is not new [2].
>
> **Response to weakness 2**:
>
> Thanks for your valuable feedback. We need to clarify that AgentRefine has two main differences from previous methods:
>
> (1) **Diversity**: Diversity is important for generalization[2], the work like KALM[3] uses a finetuned LLM to generate new trajectories based on a certain environment (physical world) and given action. So it **not only can't be used in the new environment** (os/web/reasoning etc.) **but also can't expand its action space**. AgentRefine uses diverse environments to train the agent, which can help the agent to be more robust instead of memorizing the pattern/preconditions/parameters.
>
> (2) **Refinement**: The refinement step is important for the LLM-based agent to generalize well. We are **the first (as far as we know) paper to synthesize the refinement step in the agent-tuning process and discuss its importance**.
>
> As a result, even though the previous work has used data synthesis in Agent domain, our work still has significant differences and innovations. We will clarify that our OOD setting means the model should adapt to **both new tasks and new environments** in the final version of the paper. Thanks for the suggestion.

---

> ### Author Response · Authors · 2024-11-17
> **Official Comment to Reviewer HUkR (2/2)**
>
> **Weakness 3**: The experimental results do not demonstrate a strong improvement over existing methods, which questions the practical impact of the proposed approach. An apple-to-apples comparison of your main results to show the advantage of the algorithm would make your results more straightforward and strong, instead of using a lot of underlined text to filter out the results where training data is sampled in the same environment as the task.
>
> **Response to weakness 3 (1/2)**:
>
>  Thanks for the suggestion. We need to emphasize that a model trained with OOD data is most likely to be worse than a model trained with IND data. For example, if a model is trained with GSM8k training set, it probably will perform well on GSM8k test set then the model is trained with other source math data like MATH. So **OOD test set result is more important than IND test set**.
>
> **Response to weakness 3 (2/2)**:
>
> To prove our method is better than previous methods, we filter out the IND data in the Agent-Flan[4] (about 672 samples in total 34440 samples are filtered out.), AgentGym [5]  (about 5350 samples in total 14485 samples are filtered out.) and retrain new models. Comparing the results of "AgentGym wo ind", "Agent-FLAN wo ind" and "AgentRefine", we can see that **AgentRefine outperforms the other two methods in all tasks**. This demonstrates that our method is a strong improvement over previous methods. We include this analysis in Appendix F. Thanks for the reviewer's suggestion!
>
> | Method            | Alfworld       |            | BabyAI        |            | SciWorld      |            | PDDL          |            | Jericho       |            |
> |-------------------|----------------|------------|---------------|------------|---------------|------------|---------------|------------|---------------|------------|
> |                   | Success        | Progress   | Success       | Progress   | Success       | Progress   | Success       | Progress   | Success       | Progress   |
> | LLaMA-3-8B-Instruct    | 22.4      | 46.1       | 45.5          | 56.5       | 7.8           | 41.1        | 10.0          | 38.4      | 0.0           | 24.3       |
> | AgentGen          | 29.1           | 47.6       | 20.5          | 35.0       | -             | -          | 11.7          | 23.0       | -             | -       |
> | AgentGym          | _61.9_          | _76.9_       | _47.3_        | _61.4_      | _18.9_       | _47.5_      | 1.7           | 16.6       | 0.0          | 12.9       |
> | AgentGym wo ind   | 5.9            | 28.7       | 27.7          | 40.0       | 2.2           | 14.3       | 8.2           | 18.8       | 5.0           | 13.7       |
> | Agent-FLAN         | _67.2_           | _79.7_      | 25.0          | 53.5       | 1.1           | 10.9       | 8.3           | 25.5       | 0.0             | 9.1        |
> | Agent-FLAN wo ind  | 1.5            | 19.7       | 32.1          | 45.0       | 2.2           | 12.1       | 6.6           | 23.6       | 0.0             | 14.5       |
> | AgentRefine       | 44.8           | 83.8       | 37.5          | 50.4       | 14.4          | 42.6       | 16.6          | 37.8       | 10.0          | 32.3       |
>
> The italic text indicates that the training data is sampled in the same environment as the task and is considered as held-in evaluation. "wo ind" means the model is trained without the IND data.
>
>
>
> [1] Changing Answer Order Can Decrease MMLU Accuracy.
>
> [2] What Makes Good Data for Alignment? A Comprehensive Study of Automatic Data Selection in Instruction Tuning
>
> [3] Knowledgeable Agents by Offline Reinforcement Learning from Large Language Model Rollouts.
>
> [4] Agent-flan: Designing data and methods of effective agent tuning for large language models
>
> [5] Agentgym: Evolving large language model-based agents across diverse environments.

---

> ### Author Response · Authors · 2024-11-30
> **A Kind Reminder for Reading the Response**
>
> Dear Reviewer HUkR,
>
> Thank you for your insightful suggestions. We have revised the paper, added 2 more perturbation experiments and a IND filtering experience. In the response above, we have also tried to clarify progresses on this research direction and our contributions over previous (concurrent) works which received approval from other reviewers. Since the rebuttal period is closing very soon, could you please check our response to see whether it mitigates your concerns? We would greatly appreciate that!
>
> Thank you for your time and consideration,
>
> The authors

---

> > ### Comment · Reviewer_HUkR · 2024-12-01
> >
> > Thank you for your response. Most of my concerns have been resolved. I have raised my score and recommend that the authors include the mentioned works and experiments in the revised paper.

---

> ### Author Response · Authors · 2024-12-01
> **Thank you for Reviewer's reply**
>
> Dear Reviewer HUkR,
>
> We are pleased to see that our response has alleviated your concerns. Your suggestions have all been very helpful to us. We will continue update our paper! Thank you again for your thorough review and suggestions regarding our work.
>
> ICLR 2025 Submission14212 Author

---

### Official Review · Reviewer_i2Mf · 2024-11-11

**Soundness:** 2
**Presentation:** 3
**Contribution:** 2
**Rating:** 6
**Confidence:** 3

**Summary:**

The paper proposes a novel framework to improve the generalization capabilities of LLMs based agents. The authors identify that existing agent-tuning methods often overfit to specific environments and fail to generalize to new tasks. To address this, the paper introduces AgentRefine, which leverages a agent synthesis framework to encompass a diverse array of environments and tasks drawing upon extensive human persona data, enabling the model to learn from its mistakes through a process of refinement tuning. The experiments demonstrate that AgentRefine method outperforms state-of-the-art methods in terms of generalization, robustness to perturbations, and the ability to generate diverse thoughts during inference.

**Strengths:**

The proposed method's idea seems like meta learning, which trains the policy on diverse tasks for quickly adapting to novel tasks. This idea makes sense to me and seems new in agent domain.

I appreciate authors' rethinking on the generalization of agent-tuning. The issue of memorizing trajectory leading to overfitting seems valid to me.

The experiment evaluates the performance of AgentRefine from wide range of perspectives.
The findings establish a correlation between agent generalization and multi-task agent training mechanism / self-refinement, providing a new paradigm for future research in agent-tuning.

**Weaknesses:**

Overall AgentRefine is a simple and effective method. However, the main idea is not new, as discussed in related work, Agent-FLAN and AgentGen have proposed to train generalist agents using general data. The idea of refinement is also widely studied as discussed in introduction. I encourage authors to clearly differentiate AgentRefine from these prior works. Highlight unique aspects or improvements over existing methods. Consider incorporating a comparative analysis to demonstrate the advantages of AgentRefine.

I feel the procedure suffers from a high risk of generating low-diversity tasks, as the script generation is based on human persona data, which is limited in a certain domain. In contrast, a generalist agent is expected to complete any tasks.

The goal of the proposed method is to build a LLM-based agent to generalize to novel tasks. However, this way to generate agent tasks does not bring new knowledge to LLMs, but enabling the LLMs to follow the output format more strictly, as it trains LLMs on the data generated by LLMs themselves.

Besides, the source of performance improvement is not clear. For instance, why the LLM-generated trajectories can improve performance on novel tasks? Authors can provide some examples of the evaluation tasks, and examples of the generated tasks.

**Questions:**

Refer to weakness section.

---

> ### Author Response · Authors · 2024-11-17
> **Official Comment to Reviewer i2Mf (1/2)**
>
> Thanks for reviewer's comments. Here are our responses to the comments.
>
> ---
>
> **Weakness 1**:  Overall AgentRefine is a simple and effective method. However, the main idea is not new, as discussed in related work, Agent-FLAN and AgentGen have proposed to train generalist agents using general data. The idea of refinement is also widely studied as discussed in introduction. I encourage authors to clearly differentiate AgentRefine from these prior works. Highlight unique aspects or improvements over existing methods. Consider incorporating a comparative analysis to demonstrate the advantages of AgentRefine.
>
> **Response to weakness 1 (1/2)**:
>
>  Apologies for the confusion. We appreciate the opportunity to clarify the differences between our work and previous agent-tuning studies. In recent years, LLM-Based Agents have become a popular paradigm. However, improving LLM performance on agent tasks during the post-training phase remains a challenging issue. Previous work typically sampled and trained in fixed and single-agent environments (with Held-in data that is distributionally similar to the test data), which significantly improved performance on specific tasks (test sets that are distributionally similar to the training data). However, performance drops sharply once the task changes.
>
> AgentTuning was the first to recognize this issue by adding a portion of general alignment data to the single-agent data, alleviating the problem and demonstrating initial generalization capabilities. Agent-Flan further improved the agent data, enhancing the model's generalization in agent tasks.
>
> In our work, we demonstrate that **the above approaches still have significant limitations** in terms of generalization, specifically in terms of easily overfitting on single data sets, stacking in reasoning, and learning incorrect reasoning patterns (as discussed in Figure 2, Figure 9 and Section 4.3, etc.). To address this issue, we **increased the diversity of training agent data** through synthetic data, significantly alleviating the model's overfitting problem. Additionally, we add refinement steps in the trajectory. We show that whether the training data **includes the refinement process** affects the model's reasoning pattern, and adding synthetic refinement processes greatly enhances the generalization performance of LLMs.
>
> **Response to weakness 1 (2/2)**:
>
> Apologies for the confusion. We appreciate the opportunity to clarify the differences between our work and previous self-refine studies[1]. Previous self-refine methods refine the output at the instance level, the refinement is compulsory. So they have 2 flaws: 1. **Refinement will be generated when the output is correct and may disturb the model's thought and create wrong output**. 2. If the refinement does not work, **it can't refine again**. However, AgentRefine refines the decision at the step level with reflection, self-correction as well as multi-path exploration. AgentRefine's refinement can be generated spontaneously instead of some prompt/pipeline strategy. This type of refinement is more natural and approaches the essence of human thinking. (as discussed in the O1 Replication Journey[2], which publiced after ICLR's submission deadline). We also need to emphasize that **AgentRefine is the first paper (as far as we know) to analyze the relationship between the refinement process and the generalization of LLM Agents.**
>
> ---
>
> **Weakness 2**: I feel the procedure suffers from a high risk of generating low-diversity tasks, as the script generation is based on human persona data, which is limited in a certain domain. In contrast, a generalist agent is expected to complete any tasks.
>
> **Response to weakness 2**:
>
>  Apologies for the confusion. Persona data [3] is a diverse and rich information content. Persona hub[3] contains 1,000,000,000 personas after filtering via diverse. If **the filter cosine similarity is 0.5, it can still generate 1 million diverse personas**. The persona hub also proved that **the data generated via the persona hub has similar diversity to the persona data**  (Figure 10 in Persona Hub) and its scaling experience (Figure 9 in Persona hub) shows that data **generated via the persona hub is not yet saturated with the size of 1M under math problem**, which is really hard for another method (because of the question diversity). So it probably **will not be limited to a certain domain**.

---

> ### Author Response · Authors · 2024-11-17
> **Official Comment to Reviewer i2Mf (2/2)**
>
> **Weakness 3**: The goal of the proposed method is to build an LLM-based agent to generalize to novel tasks. However, this way of generating agent tasks does not bring new knowledge to LLMs but enables the LLMs to follow the output format more strictly, as it trains LLMs on the data generated by LLMs themselves.
>
> **Response to weakness 3**:
>
>  Apologies for the confusion. We do believe that training LLMs can **not only bring new knowledge** to LLMs but also **improve the LLMs' ability** to solve the task. For example, when training the math problem, the model can learn the reasoning ability via COT and it does not "bring new knowledge". AgentRefine can help the model to get self-correction, reflection, and multi-path exploration abilities, which are important for the model to solve the task.
>
> ---
>
> **Weakness 4**:  Besides, the source of performance improvement is not clear. For instance, why the LLM-generated trajectories can improve performance on novel tasks? Authors can provide some examples of the evaluation tasks, and examples of the generated tasks.
>
> **Response to weakness 4 (1/2)**:
>
>  We believe there are 2 following reasons that can improve performance on novel tasks in our method:
>
> 1. Environment diversity: By using LLM-generated environments to create the LLM-generated trajectories, the task and environment are diverse (as discussed in Response to weakness 2), which can help the model to be more robust and generalization. Other works like Agent-Flan[4] and AgentGym[5] use a small number of human-labeled environments, which may not be diverse enough.
>
> 2. Refinement: Our trajectories contain the refinement step, which can teach the model to learn self-correction, reflection, and multi-path exploration abilities. This doesn't happen in other agent-tuning methods.
>
> **Response to weakness 4 (2/2)**:
>
>  Apologies for the confusion. We provide examples of the evaluation tasks in Figure 9 and examples of the generated tasks in Supplementary Material. They show the importance of diversity and refinement! Thanks for the valuable consideration.
>
>
>
> [1] Self-refine: Iterative refinement with self-feedback.
>
> [2] O1 Replication Journey: A Strategic Progress Report -- Part 1
>
> [3] Scaling Synthetic Data Creation with 1,000,000,000 Personas
>
> [4] Agent-flan: Designing data and methods of effective agent tuning for large language models
>
> [5] Agentgym: Evolving large language model-based agents across diverse environments.

---

> ### Comment · Reviewer_i2Mf · 2024-11-17
>
> Thanks for authors' response. I read authors' rebuttal and other reviewers' comments carefully. Major concerns lie in the method's novelty, experimental evaluation and the usage of GPT-4.
>
> Regarding the novelty, I like the method that uses extensive persona files to generate scripts for trajectory generation (but I am not sure whether it is new in the community). It seems like a promising way to create diverse demonstrations for better generalization.
>
> Regarding the usage of GPT-4, I realize the new knowledge may come from this stronger model that generates new trajectories (as authors' discussions with reviewer opnM). As authors' response regarding the source of new knowledge is still unclear, I would like to hear authors' further comments on these points.

---

> ### Author Response · Authors · 2024-11-17
> **Official Comment to Reviewer i2Mf**
>
> Thanks for your affirmation of the novelty in this paper.
>
> **Question 1**:  Authors' response regarding the source of new knowledge is still unclear, I would like to hear authors' further comments on these points.
>
> Response to question 1: Thanks for the suggestion. As we mentioned in "Response to weakness 3": **We do believe that training LLMs can not only bring new knowledge to LLMs but also improve the LLMs' ability to solve the task.** If you believe that the ability is also a kind of knowledge, your claim is right, the new knowledge may come from the stronger model (**GPT-4, Deepseek-v2.5** (Deepseek's experiment is in Section 5) etc.) that generates new trajectories.
>
>  Specifically, the new knowledge may come from:
>
> (1) Reasoning and planning knowledge: reasoning and planning step (most agent-tuning data have), the reflection step, the self-correction step, and the multi-path exploration step (only in AgentRefine) in the trajectory.
>
> (2) Insturaction following knowledge: The action format and output format in the trajectory.
>
> (3) Long-context knowledge:  Multi-turn trajectory and the refinement step which uses the information in the observation several turns before
>
> We also need to emphasize that we are **the first (as far as we know) paper that finds the refinement knowledge (ability) is important for the LLM Agent to generalize well**.
>
> If you have any further questions, please feel free to ask.
>
> Thank you for your reply!

---

> ### Comment · Reviewer_i2Mf · 2024-11-24
>
> Thanks for your response. I feel more positive about the paper after reading the author feedback. Thus I have raised my score. The major weakness lies in the rely on stronger model to generate trajectories, and the uncontrollable performance on a specific task by training on extensive generated tasks.

---

> ### Author Response · Authors · 2024-12-01
> **Thank you for Reviewer's reply**
>
> Dear Reviewer HUkR,
>
> We are pleased to see that our response has alleviated your concerns. Your suggestions have all been very helpful to us. Thank you again for your thorough review and suggestions regarding our work.
>
> ICLR 2025 Submission14212 Author

---

### Author Response · Authors · 2024-11-21
**General Response to Reviewers and Revision Submitted**

We thank all the reviewers for their insightful comments and suggestions. We have revised the paper to address the reviewers’ concerns. Below we summarize the major revisions (the main revisions are marked with blue text in the pdf, we also made some minor layout changes to fit the page limit), while we reply to the comments of each reviewer separately.

The major revisions are:

1. Add an explanation to clarify the differences between our work and previous agent-tuning studies in Appendix B and the related work. (Reviewer i2Mf, HUkR)
2. Revise the introduction of self-refine in the related work. (Reviewer HUkR)
3. Provide an introduction to the persona hub in Appendix C. (Reviewer i2Mf,opnM)
4. Add IND filtering experiment to eliminate the influence of IND dataset in Appendix F. (Reviewer HUkR)
5. Add a new method - Reflexion in Appendix G. (Reviewer NknQ)
6. Add a new reasoning benchmark - HotpotQA in Section 6. (Reviewer NknQ)
7. Provide the standard deviation of the results in Appendix H. (Reviewer NknQ)
8. Provide an explanation to clarify the inference pipeline in the main text and updated Figure 4. (Reviewer NknQ, opnM)
9. Add a GPT-4 judgement verification experiment in Section 8. (Reviewer opnM)
10. Add 2 more perturbations to the robustness experiment, analyze the contribution among different components and update perturbation details in the Section4.3, Appendix I and Appendix K. (Reviewer opnM)
11. Correct the typos and grammar mistakes. (Reviewer NknQ)
12. Move Section "Synthesis from Open Source Model" from Appendix F to Section 5.  (Reviewer opnM)

We appreciate the reviewers for their valuable comments and suggestions.

---

### Comment · Area_Chair_JSwU · 2024-11-25

Dear Reviewers,


This is a friendly reminder that the discussion will end on Nov. 26th (anywhere on Earth). If you have not already, please take a close look at all reviews and author responses, and comment on whether your original rating stands.


Thanks,

AC

---

### Meta-Review · Area_Chair_JSwU · 2024-12-21

**Metareview:**

The paper introduces AgentRefine, a novel framework aimed at enhancing the generalization capabilities of large language model (LLM)-based agents. The approach tackles the overfitting problem prevalent in existing agent-tuning methods by using a data generation pipeline that simulates diverse environments and tasks.
- The framework avoids task-specific overfitting by synthesizing data with minimal reliance on task-specific information.
- The method addresses the generalization gap by leveraging diverse environments and tasks, ensuring agents adapt well to held-out scenarios.
- Experimental results demonstrate that AgentRefine outperforms existing baselines, highlighting its effectiveness.

The weaknesses are (1)the main idea, while practical, is not significantly novel; (2) the method heavily depends on GPT-4 for script and trajectory generation as well as error verification.

Currently, I think the strengths outweigh the weaknesses.

**Additional Comments On Reviewer Discussion:**

During rebuttal, more tasks and baselines are added, which address many of the reviewers' concerns. There are still remaining concerns, i.e., reviewer opnM is still negative about this paper and has raised the score to 5.

This is a borderline paper.

---

### Decision · Program_Chairs · 2025-01-22

Accept (Poster)